# Alignment through Meta-Weighted Online Sampling: Bridging the Gap between Data Generation and Preference Optimization

**Junming Yang[1,2], Ning Xu[1,2,*], Biao Liu[1,2], Shiqi Qiao[1,2], Xin Geng[1,2]**
[1]School of Computer Science and Engineering, Southeast University, Nanjing, China
[2]Key Laboratory of New Generation Artificial Intelligence Technology and
 Its Interdisciplinary Applications (Southeast University), Ministry of Education, China
`{jmingyang,xning,liubiao01,sqqiao,xgeng}@seu.edu.cn`

## Abstract

Preference optimization is crucial for aligning large language models (LLMs) with human values and intentions. A significant challenge in this process is the distribution mismatch between pre-collected offline preference data and the evolving model policy. Existing methods attempt to reduce this gap using static heuristics or decoupled online sampling strategies, but they often fail to adapt to the model's dynamic learning state. To bridge this gap, we propose Meta-Weighted Adaptive Preference Optimization (MetaAPO), a novel framework that dynamically couples data generation with model training. MetaAPO employs a lightweight meta-learner, as an "alignment gap estimator", to evaluate the potential benefits of on-policy sampling in relation to offline data. This guides targeted online generation and assigns sample-wise meta-weights to the optimization objective, dynamically balancing the quality and distribution of online and offline data. Experiments on AlpacaEval 2, Arena-Hard and MT-Bench demonstrate that MetaAPO consistently outperforms existing preference optimization approaches across various settings, while reducing 42% in online annotation costs. Code is available at https://github.com/junming-yang/MetaAPO.

## 1 Introduction

Learning from human feedback is essential for aligning large language models (LLMs) with human values and intentions (Ouyang et al., 2022), ensuring they are helpful, honest and harmless (Achiam et al., 2023; Grattafiori et al., 2024). This advancement is primarily achieved in the model alignment stage, which involves training reward models or LLMs directly on datasets curated from human preferences. Reinforcement learning from human feedback (RLHF) (Stiennon et al., 2020; Ouyang et al., 2022) is a widely used method for fine-tuning LLMs to achieve this goal. However, RLHF often faces challenges, particularly in computational efficiency and training stability. Recently, alternative offline algorithms like DPO (Rafailov et al., 2023), SimPO (Meng et al., 2024) and KTO (Ethayarajh et al., 2024) have been explored. These methods reparameterize the RLHF objective to directly optimize the policy using pre-collected offline preference datasets, simplifying the training process and improving efficiency.

As alignment research progresses, recent studies have emphasized that the quality and distribution of preference data are critical for effective model alignment (Zhou et al., 2023; Deng et al., 2025; Xia et al., 2024; Ding et al., 2024). Although offline preference datasets are diverse and scalable, they often suffer from a distribution mismatch between the pre-collected training data and current policy, as the offline data is usually generated by a different model. This mismatch leads to out-of-distribution (OOD) issues that hinder preference alignment performance (Yan et al., 2024; Xu et al., 2024; Tajwar et al., 2024). To improve the quality and distribution of training data, existing methods for mitigating distribution shifts generally fall into two categories. Some methods improve the reliability of offline data by filtering out low-quality or OOD samples using manually designed metrics, such as average

---

*Corresponding author

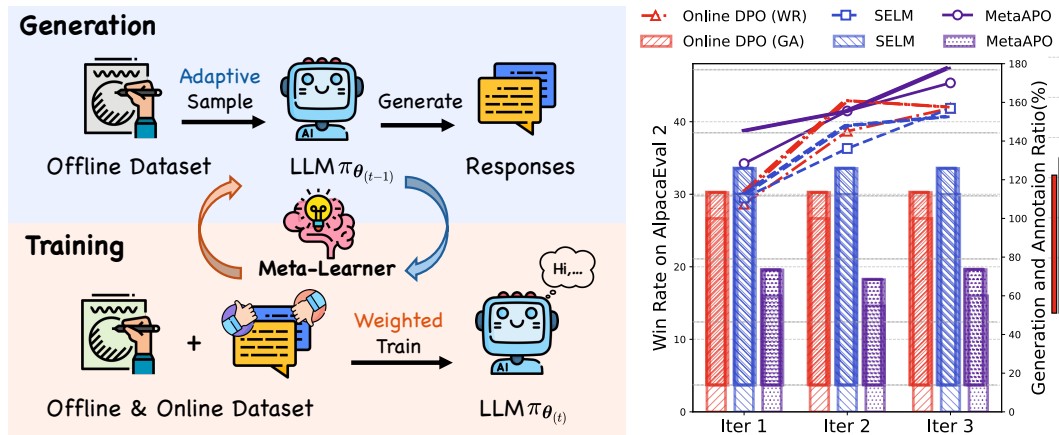

Figure 1: **Left:** Overview of MetaAPO. MetaAPO employs a meta-learner to couple online data generation (top) and model training (bottom). The meta-learner adaptively assigns weights by evaluating offline data, guiding both targeted online sample generation and training on the weighted combination of offline and online samples. **Right:** Performance (left y-axis, line plots) and online generation and annotation ratio relative to Online DPO (right y-axis, bar plots) across training iterations for different methods.

validation loss, as exemplified by Selective DPO (Gao et al., 2025). Others attempt to mitigate distribution shifts by constructing training data through online sampling (Zhang et al., 2024b; Ko et al., 2024; Badrinath et al., 2024). For instance, Iterative DPO (Xiong et al., 2023; Guo et al., 2024b) and SPPO (Wu et al., 2024) follow an iterative sampling-alignment loop: the current policy first generates responses, which are then annotated by a reward model to collect preference data for direct preference alignment. The updated policy is subsequently used to repeat this process.

While human-annotated offline data typically offers high efficiency and quality, the distribution mismatch between static datasets and the evolving model policy can negatively impact alignment performance. On the other hand, model-generated data obtained through online sampling more accurately reflects the current policy distribution, which helps address the distribution shifts. However, online data often lacks diversity and quality due to its reliance on the model's capabilities and current alignment state, leading to preference data that may be inefficient or even noisy. This trade-off underscores the need for adaptive mechanisms that can dynamically leverage the efficiency and diversity of offline data while taking advantage of the distribution benefits of online data. However, existing methods primarily depend on heuristic, manually designed metrics (*e.g.,* static thresholds) for data selection and generation (Dong et al., 2024; Gao et al., 2025; Zhang et al., 2024b; Ji et al., 2024), overlooking the interaction between the data sampling process and preference optimization. An integrated approach that simultaneously accounts for data quality, distribution alignment, and the dynamics of preference optimization is essential for more effective alignment.

Motivated by the above considerations, we propose Meta-Weighted Adaptive Preference Optimization (MetaAPO), a novel framework that closely couples data sampling with model alignment through a lightweight, learnable meta-learner. As demonstrated in Figure 1 (Left), the meta-learner acts as an "alignment gap estimator", predicting the potential improvements that online generations can provide relative to offline data. This enables targeted online sampling, prioritizing prompts expected to yield the highest alignment gains. In addition, the meta-learner adjusts the training process by assigning sample-wise meta-weights to a hybrid loss function, adaptively balancing the contributions of offline data and online samples according to their respective quality and distribution advantages. This dynamic interaction ensures that the model continually adapts its learning focus to mitigate distribution mismatches while avoiding redundant online sampling for already aligned data, thus improving both training efficiency and performance. Empirical results demonstrate that MetaAPO significantly improves alignment across various base models, outperforming both offline and online approaches. Our contributions can be summarized as follows:

- We propose MetaAPO, an extensible alignment framework compatible with various preference optimization methods. It bridges the gap between online data generation and model training by adaptively coupling these two stages, enabling a model-aware alignment process.

- Our method employs a meta-learner that adaptively assigns sample-wise weights to data. These weights serve two purposes: guiding online data sampling based on the model's training needs to mitigate distribution mismatches and adjusting the training by emphasizing beneficial samples.
- Extensive experiments demonstrate that MetaAPO achieves superior alignment performance compared to baselines, even reducing 42% online annotation requirements (Figure 1, Right).

## 2 RELATED WORK

**Alignment with Preference Data.** RLHF is a widely used method for aligning LLMs with human values and intentions (Stiennon et al., 2020; Ouyang et al., 2022; Bai et al., 2022). While effective, RLHF suffers from high computational costs and training instability, prompting the exploration of alternative alignment strategies. One prominent direction is offline preference alignment (Azar et al., 2024; Dong et al., 2023; Yuan et al., 2023; Lu et al., 2024), which directly trains models using pre-collected datasets of human preferences, such as DPO (Rafailov et al., 2023), SimPO (Meng et al., 2024) and KTO (Ethayarajh et al., 2024). These methods are attractive for their efficiency and simplicity. However, a key limitation is the distribution mismatch between static training data and the evolving outputs of the policy model, which often leads to poor generalization on out-of-distribution responses (Xu et al., 2024; Yan et al., 2024; Yang et al., 2025; Tajwar et al., 2024). To overcome this limitation, online preference alignment methods such as Iterative DPO (Xiong et al., 2023) and SPPO (Wu et al., 2024) have been proposed. These approaches adopt an iterative training paradigm in which each round involves generating new responses from the current policy and optimizing the model using annotated preferences (Yuan et al., 2024; Ko et al., 2024; Chen et al., 2024; Kou et al., 2025). By enabling exploration beyond the limitations of the initial static dataset, they can improve model performance (Su et al., 2025; Xu et al., 2024; Tajwar et al., 2024). However, they primarily rely on on-policy data and often overlook valuable offline preference datasets. Bridging both paradigms, hybrid methods like SELM (Zhang et al., 2024b) integrate offline and online data by using a reward model to compare responses from both sources, thereby constructing more diverse preference sets. Nonetheless, these hybrid approaches lack strategic sampling and generation mechanisms, failing to account for the distribution relationship between offline and online data, which leads to inefficient resource utilization and suboptimal performance.

**Preference Data Selection and Augmentation.** These limitations underscore the growing importance of strategic data sampling and targeted augmentation in preference alignment. The "less is more" principle suggests that selecting high-quality and relevant data often leads to better outcomes than relying on large volumes of unfiltered data (Zhou et al., 2023; Deng et al., 2025; Liu et al., 2023; Ding et al., 2024). Consequently, various data selection methods aim to identify the most informative and helpful samples from general-purpose datasets to improve training effectiveness (Wang et al., 2025; Khaki et al., 2024; Badrinath et al., 2024). For instance, Selective DPO (Gao et al., 2025) employs loss-based filtering to discard uninformative or harmful samples, while MAP (Huang et al., 2025) uses internal and external reward margins to guide sample inclusion. Beyond sample selection, data augmentation further enhance alignment (Muldrew et al., 2024; Gou & Nguyen, 2024; Guo et al., 2024a; Liu et al., 2024; Das et al., 2024; Liu et al., 2025). ADPO (Ji et al., 2024) filters offline data based on a manual threshold and then performs online augmentation on the selected samples. The Reward-Augmented method (Zhang et al., 2024a) augments the dataset by reversing preference pairs under the guidance of prompts. DICE (Chen et al., 2024) and SeRA (Ko et al., 2024) combine data augmentation with filtering methods, guided by length regularization and reward margins to enhance training data. Despite these advancements, most prior work treats data sampling as an auxiliary step, often decoupled from the optimization process and guided by static heuristics. In contrast, MetaAPO proposes a framework that closely integrates online data generation with preference optimization via adaptive meta-weighting, mitigating the distribution mismatch in preference alignment.

## 3 PRELIMINARIES

**Reinforcement Learning from Human Feedback.** RLHF aims to align the language model policy $\pi_{\boldsymbol{\theta}}(\cdot|\boldsymbol{x})$ with human preferences over prompts $\boldsymbol{x} \in \mathcal{X}$. A typical RLHF pipeline (Ouyang et al., 2022) starts by training a reward model on a preference dataset $\mathcal{D} = \{(\boldsymbol{x}, \boldsymbol{y}_w, \boldsymbol{y}_l)^i\}_{i=1}^{N}$, where each preference pair is annotated such that $\boldsymbol{y}_w \succ \boldsymbol{y}_l$ indicates that $\boldsymbol{y}_w$ is preferred over $\boldsymbol{y}_l$ for prompt $\boldsymbol{x}$. The reward model $r_{\psi}(\boldsymbol{x}, \boldsymbol{y})$ is trained using the Bradley–Terry (BT) model (Bradley & Terry, 1952),

by minimizing a negative log-likelihood loss that encourages larger reward gaps between preferred and dispreferred responses:

$$\mathcal{L}_{\text{RM}}(\boldsymbol{\psi}) = -\mathbb{E}_{(\boldsymbol{x}, \boldsymbol{y_w}, \boldsymbol{y_l}) \sim \mathcal{D}} \left[ \sigma \left( r_{\boldsymbol{\psi}}(\boldsymbol{x}, \boldsymbol{y}_w) - r_{\boldsymbol{\psi}}(\boldsymbol{x}, \boldsymbol{y}_l) \right) \right], \tag{1}$$

where $\sigma(\cdot)$ denotes the logistic function. The policy $\pi_{\boldsymbol{\theta}}$ is then optimized via reinforcement learning to maximize the expected reward while staying close to a reference policy $\pi_{\text{ref}}$ :

$$\max_{\pi_{\boldsymbol{\theta}}} \mathbb{E}_{\boldsymbol{x} \sim \mathcal{D}, \boldsymbol{y} \sim \pi_{\boldsymbol{\theta}}(\cdot|\boldsymbol{x})}[r_{\boldsymbol{\psi}}(\boldsymbol{x}, \boldsymbol{y})] - \beta \mathbb{D}_{\text{KL}}[\pi_{\boldsymbol{\theta}}(\boldsymbol{y}|\boldsymbol{x})||\pi_{\text{ref}}(\boldsymbol{y}|\boldsymbol{x})], \tag{2}$$

where $\beta$ controls the trade-off between reward maximization and deviation from the reference policy.

**Direct Preference Optimization.** While effective, RLHF can be complex, unstable, and computationally demanding. DPO (Rafailov et al., 2023) simplifies the process by directly optimizing the policy using a supervised learning objective over preference data, without explicitly learning a reward model. DPO reparameterizes the reward as $r(\boldsymbol{x}, \boldsymbol{y}) = \beta(\log \frac{\pi_{\boldsymbol{\theta}}(\boldsymbol{y}|\boldsymbol{x})}{\pi_{\text{ref}}(\boldsymbol{y}|\boldsymbol{x})} + \log Z(\boldsymbol{x}))$, where $Z(\boldsymbol{x})$ is the partition function and $\pi_{\text{ref}}$ is typically the supervised fine-tuned (SFT) model. By minimizing the negative log-likelihood of reward modeling in Eq. 1, DPO derives the following loss:

$$\mathcal{L}_{\text{DPO}}(\boldsymbol{\theta}) = -\mathbb{E}_{(\boldsymbol{x}, \boldsymbol{y_w}, \boldsymbol{y_l}) \sim \mathcal{D}} \left[ \log \sigma \left( \beta \log \frac{\pi_{\boldsymbol{\theta}}(\boldsymbol{y}_w|\boldsymbol{x})}{\pi_{\text{ref}}(\boldsymbol{y}_w|\boldsymbol{x})} - \beta \log \frac{\pi_{\boldsymbol{\theta}}(\boldsymbol{y}_l|\boldsymbol{x})}{\pi_{\text{ref}}(\boldsymbol{y}_l|\boldsymbol{x})} \right) \right]. \tag{3}$$

DPO supports both offline preference datasets $\mathcal{D}_{\text{offline}}$ (*e.g.*, curated in advance) and online datasets $\mathcal{D}_{\text{online}}$ (generated by the policy model $\pi_{\boldsymbol{\theta}}$).

## 4 METHOD

In this section, we present the technical details of MetaAPO. The alignment process is conducted within a single training epoch, where the dataset is divided into sequential subsets and processed iteratively over $t \in \{0, 1, \dots, T\}$. In each iteration $t$, we start by performing meta-weighted adaptive online sampling to generate online data from policy $\pi_{\boldsymbol{\theta}_{(t-1)}}$ (Section 4.1). The policy is then fine-tuned on the hybrid dataset using the meta-weighted objective to obtain the updated policy $\pi_{\boldsymbol{\theta}_{(t)}}$ (Section 4.2). Concurrently, the meta-learner is periodically updated to better estimate the potential gain from online data (refer to Section 4.3). The overall workflow is illustrated in Algorithm 1.

### 4.1 META-WEIGHTED ADAPTIVE ONLINE SAMPLING

We begin with a model-aware sampling strategy for acquiring new preference data. This process involves using a preference scoring function (derived from objectives like DPO) to assess the utility of existing offline samples. Given a prompt $\boldsymbol{x}$ and a pair of preferred and dispreferred responses $(\boldsymbol{y}_w, \boldsymbol{y}_l)$, the instance-level preference score is computed as:

$$\ell(\boldsymbol{x}, \boldsymbol{y}_w, \boldsymbol{y}_l) = \log \sigma \left( \beta \log \frac{\pi_{\boldsymbol{\theta}}(\boldsymbol{y}_w|\boldsymbol{x})}{\pi_{\text{ref}}(\boldsymbol{y}_w|\boldsymbol{x})} - \beta \log \frac{\pi_{\boldsymbol{\theta}}(\boldsymbol{y}_l|\boldsymbol{x})}{\pi_{\text{ref}}(\boldsymbol{y}_l|\boldsymbol{x})} \right), \tag{4}$$

where a higher score $\ell(\cdot)$ indicates stronger agreement between the current policy $\pi_{\boldsymbol{\theta}}$ and human preferences, compared to a fixed reference policy $\pi_{\text{ref}}$. This scoring function is applied to each instance in the offline preference dataset $\mathcal{D}_{\text{offline}} = \{(\boldsymbol{x}, \boldsymbol{y}_w^{\text{off}}, \boldsymbol{y}_l^{\text{off}})^i\}_{i=1}^N$, producing scores $\ell^{\text{off}}(\boldsymbol{x}, \boldsymbol{y}_w^{\text{off}}, \boldsymbol{y}_l^{\text{off}})$.

To enable the sampling process to dynamically adapt to the model's current state and requirement, we introduce a meta-learner that adaptively re-weights each offline sample based on its preference score. The meta-learner $h_{\boldsymbol{\phi}}$, implemented as a two-layer MLP, maps the preference score $\ell(\cdot)$ to a weight $w = h_{\boldsymbol{\phi}}(\ell^{\text{off}}(\cdot)) \in [0, 1]$ (Line 7 in Alg. 1). A high weight indicates that the sample aligns well with the model's current behavior, while a low weight suggests potential misalignment that may require further online exploration. This formulation can be extended to the multi-objective setting (Zhou et al., 2024). Suppose we consider $M$ objectives (*e.g.,* helpfulness, harmlessness, and honesty), each inducing its own partial preference relation and corresponding offline preference loss $\ell_i^{\text{off}}(\cdot)$, where $i \in \{1, ..., M\}$. We then apply objective-wise label enhancement (Xu et al., 2019; 2023) by assigning an adaptive continuous label degree for each objective: $d_i = h_{\boldsymbol{\phi}}(\ell_i^{\text{off}}(\cdot)) \in [0, 1], i \in \{1, ..., M\}$. The normalized label distribution (Geng, 2016) $\mathbf{d} = [\hat{d}_1, \dots, \hat{d}_m]$, derived from the initial vector $[d_1, \dots, d_m]$, characterizes the model's relative alignment-aware importance across different objectives.

---

**Algorithm 1** Training Scheme of MetaAPO

---

1: **Input:** Offline preference dataset $\mathcal{D}_{\text{offline}} = \{(\boldsymbol{x}, \boldsymbol{y}_w^{\text{off}}, \boldsymbol{y}_l^{\text{off}})^i\}_{i=1}^N$;
2:        Language model $\pi_{\boldsymbol{\theta}}$, reference model $\pi_{\text{ref}}$, reward model $\mathcal{R}$, meta-learner $h_{\boldsymbol{\phi}}$
3: **Initialize:** Load model parameters $\boldsymbol{\theta}$, initialize meta-learner parameters $\boldsymbol{\phi}$
4: **for** each iteration **do**
5:      Initialize augmentation dataset $\mathcal{D}_{\text{aug}} \leftarrow \emptyset$ and meta buffer $\mathcal{B}_{\text{meta}} \leftarrow \emptyset$
6:      **for** each $(\boldsymbol{x}, \boldsymbol{y}_w^{\text{off}}, \boldsymbol{y}_l^{\text{off}})^i \in \mathcal{D}_{\text{offline}}$ **do**     ▷ **Meta-Weighted Adaptive Online Sampling**
7:         Compute meta-weight: $w_i \leftarrow h_{\boldsymbol{\phi}}(\ell(\boldsymbol{x}, \boldsymbol{y}_w^{\text{off}}, \boldsymbol{y}_l^{\text{off}}))$
8:         Sample $u_i \sim \text{Uniform}(0, 1)$
9:         **if** $u_i > w_i$ **then**
10:           Generate $K$ responses: $\{\boldsymbol{y}_1^{\text{on}}, \dots, \boldsymbol{y}_K^{\text{on}}\} \leftarrow \pi_{\boldsymbol{\theta}}(\cdot \mid \boldsymbol{x})$
11:           Evaluate with $\mathcal{R}$: $\{\boldsymbol{y}_l^{\text{on}}, \boldsymbol{y}_w^{\text{on}}\} \leftarrow \mathcal{R}(\boldsymbol{x}, \{\boldsymbol{y}_j^{\text{on}}\}_{j=1}^K)$
12:           Add to augmentation set: $\mathcal{D}_{\text{aug}} \leftarrow \mathcal{D}_{\text{aug}} \cup \{(\boldsymbol{x}, \boldsymbol{y}_w^{\text{off}}, \boldsymbol{y}_l^{\text{off}}, \boldsymbol{y}_l^{\text{on}}, \boldsymbol{y}_w^{\text{on}})^i\}$
13:         **end if**
14:      **end for**
15:      **for** each batch $\mathcal{D}_{\text{batch}} \sim \mathcal{D}_{\text{aug}}$ **do**        ▷ **Meta-Weighted Model Training** (freeze $h_{\boldsymbol{\phi}}$)
16:         Compute sample-wise meta-weight: $w_j \leftarrow h_{\boldsymbol{\phi}}(\ell(\boldsymbol{x}, \boldsymbol{y}_w^{\text{off}}, \boldsymbol{y}_l^{\text{off}}))$
17:         Update language model by Eq. 5: $\boldsymbol{\theta} \leftarrow \boldsymbol{\theta} - \alpha \nabla_{\boldsymbol{\theta}} \mathcal{L}(\boldsymbol{\theta})$
18:         Add batch data to meta buffer: $\mathcal{B}_{\text{meta}} \leftarrow \mathcal{B}_{\text{meta}} \cup \mathcal{D}_{\text{batch}}$
19:         **if** batch iteration $\% \, T_{\text{meta}} = 0$ **then**        ▷ **Meta-Learner Update** (freeze $\pi_{\boldsymbol{\theta}}$)
20:           Update meta-learner by Eq. 6: $\boldsymbol{\phi} \leftarrow \boldsymbol{\phi} - \eta \nabla_{\boldsymbol{\phi}} \mathcal{L}_{\text{meta}}(\boldsymbol{\phi}; \mathcal{B}_{\text{meta}})$
21:           Reset meta buffer: $\mathcal{B}_{\text{meta}} \leftarrow \emptyset$
22:         **end if**
23:      **end for**
24: **end for**

---

This adaptive meta-weight $w$ subsequently guides the online sampling and generation phase. (Line 8-12 in Alg. 1). For each offline tuple $\{\boldsymbol{x}, \boldsymbol{y}_w^{\text{off}}, \boldsymbol{y}_l^{\text{off}}\}^i$, a value $u_i$ is sampled from $\text{Uniform}(0, 1)$. If $u_i > w_i$, the current policy $\pi_{\boldsymbol{\theta}}$ generates $K$ candidate responses $\{\boldsymbol{y}_1^{\text{on}}, \dots, \boldsymbol{y}_K^{\text{on}}\}$ for prompt $\boldsymbol{x}$. These responses are then ranked by an external reward model to produce online preference pairs $\boldsymbol{y}_{\text{pair}}^{\text{on}} = (\boldsymbol{y}_w^{\text{on}}, \boldsymbol{y}_l^{\text{on}})$. Finally, both the sampled offline and corresponding online preferences are merged into an augmentation set $\mathcal{D}_{\text{aug}}$. This augmented dataset forms a dynamic training stream that evolves with the model, enabling the model to better attend to regions where alignment gaps may exist.

## 4.2 META-WEIGHTED PREFERENCE OPTIMIZATION

During preference optimization, we employ the sample-wise meta-weight $w$ to dynamically balance the influence of offline preference data and online model-generated samples (Line 16-17 in Alg. 1). The overall objective is to minimize the meta-weighted preference loss:

$$\mathcal{L}(\boldsymbol{\theta}) = -\mathbb{E}_{(\boldsymbol{x}, \boldsymbol{y}_{\text{pair}}^{\text{off}}, \boldsymbol{y}_{\text{pair}}^{\text{on}}) \sim \mathcal{D}_{\text{aug}}} \Big[ \underbrace{w \cdot \ell_{\boldsymbol{\theta}}(\boldsymbol{x}, \boldsymbol{y}_w^{\text{off}}, \boldsymbol{y}_l^{\text{off}})}_{\text{adaptive offline loss}} + \underbrace{(1 - w) \cdot \ell_{\boldsymbol{\theta}}(\boldsymbol{x}, \boldsymbol{y}_w^{\text{on}}, \boldsymbol{y}_l^{\text{on}})}_{\text{adaptive online loss}} \Big], \quad (5)$$

where $\ell_{\boldsymbol{\theta}}(\cdot)$ denotes a standard preference alignment objective (*e.g.,* DPO in Eq. 4 or SimPO) used to align the current policy $\pi_{\boldsymbol{\theta}}$ with the target preferences. The first term captures contributions from offline preferences, while the second focuses on online model-generated preference samples.

This joint optimization enriches data diversity. The balance coefficient $w = h_{\boldsymbol{\phi}}(\ell^{\text{off}}(\cdot))$ is predicted by the meta-learner and indicates the confidence in offline preference data. When the model's behavior aligns well with the offline preference, the meta-learner assigns a higher weight $w$ to reinforce stable learning from reliable offline human-labeled examples. Conversely, when misalignment is detected, $w$ is reduced, prompting greater reliance on adaptive corrections from online feedback.

## 4.3 LEARNING TO WEIGHT: META-LEARNER FOR ALIGNMENT GAP ESTIMATION

The effectiveness of both adaptive online sampling (Section 4.1) and meta-weighted preference optimization (Section 4.2) critically depends on the quality of the dynamically assigned weights

$w = h_\phi(\ell^{\text{off}}(\cdot))$. Instead of relying on pre-defined heuristics or static weights, we propose to learn the weighting function $h_\phi$ through a meta-learning approach. The goal is for $h_\phi$ to act as an "alignment gap estimator", dynamically assessing the utility of each offline sample relative to potential online preference data, and assigning weights that effectively guide the alignment process. This allows training data to be weighted based on the model's current state, enabling continuous adaptation throughout the alignment process.

Specifically, the meta-learner $h_\phi$ is trained in an alternating manner with the policy model $\pi_\theta$. At intervals specified by a hyperparameter $T_{\text{meta}}$, the parameters of $\pi_\theta$ are temporarily frozen, and $h_\phi$ is updated using data stored in the meta buffer $\mathcal{B}_{\text{meta}}$ (Lines 19–21 in Alg. 1). This buffer aggregates preference data from multiple recent training batches of $\pi_\theta$, enabling the meta-learner to derive a more stable and generalized weighting strategy from accumulated experience. The meta-learner $h_\phi$ is optimized by minimizing the following loss:

$$\mathcal{L}_{\text{meta}}(\phi) = -\mathbb{E}_{(\boldsymbol{x}, \boldsymbol{y}^{\text{off}}_{\text{pair}}, \boldsymbol{y}^{\text{on}}_{\text{pair}}) \sim \mathcal{B}_{\text{meta}}} \Big[ \underbrace{h_\phi(\ell^{\text{off}}(\cdot)) \cdot \ell(\boldsymbol{x}, \boldsymbol{y}^{\text{off}}_w, \boldsymbol{y}^{\text{off}}_l)}_{\text{adaptive offline loss}} + \underbrace{(1 - h_\phi(\ell^{\text{off}}(\cdot))) \cdot \ell(\boldsymbol{x}, \boldsymbol{y}^{\text{on}}_w, \boldsymbol{y}^{\text{on}}_l)}_{\text{adaptive online loss}} \Big],$$

(6)

where $\ell(\boldsymbol{x}, \boldsymbol{y}^{\text{off}}_w, \boldsymbol{y}^{\text{off}}_l)$ (abbreviated as $\ell^{\text{off}}(\cdot)$) and $\ell(\boldsymbol{x}, \boldsymbol{y}^{\text{on}}_w, \boldsymbol{y}^{\text{on}}_l)$ (abbreviated as $\ell^{\text{on}}(\cdot)$) denote the preference scores for the offline and online response pairs, respectively, for the same prompt $\boldsymbol{x}$.

**What does meta-learner update do?** For a mechanistic understanding of meta-learner, it is useful to analyze the gradient of the loss function $\mathcal{L}_{\text{meta}}(\phi)$. The gradient with respect to the meta-learner parameters $\phi$ can be written as:

$$\nabla_\phi \mathcal{L}_{\text{meta}}(\phi) = \mathbb{E}_{\mathcal{B}_{\text{meta}}} \Big[ \big( \ell^{\text{on}}(\cdot) - \ell^{\text{off}}(\cdot) \big) \cdot \nabla_\phi(h_\phi(\ell^{\text{off}}(\cdot)) \Big]. \tag{7}$$

This gradient drives the adaptation of the weighting function $h_\phi$. When online samples yield higher preference scores than offline samples (*i.e.,* $\ell^{\text{on}}(\cdot) > \ell^{\text{off}}(\cdot)$), the meta-learner reduces the weight assigned to offline data, encouraging exploration guided by online feedback. Conversely, when offline preferences remain superior, their weights are increased to reinforce stable learning from reliable human annotations. The full gradient derivation is provided in Appendix A.1.

Moreover, we provide a theoretical guarantee for the meta-learner $h_\phi$ in Theorem 1. The result demonstrates that the risk of the learned weighting function is provably close to the oracle risk, with the alignment gap controlled by the hypothesis complexity and meta-buffer size.

**Theorem 1** (Generalization Bound for Meta-Learner). *Let $\hat{h}_\phi$ denote the meta-learner function learned by minimizing the empirical meta-risk over the meta-buffer $\mathcal{B}_{meta}$ of size $m$, and let $h^*$ be the oracle function that minimizes the true meta-risk over the hypothesis space $\mathcal{H}$. Assume the meta-loss $\mathcal{L}_{meta}(\phi)$ is bounded in $[0, M]$ for any sample $(\boldsymbol{x}, \boldsymbol{y}^{off}_{pair}, \boldsymbol{y}^{on}_{pair})$. Then, for any $\delta > 0$, with probability at least $1 - \delta$, the following inequality holds:*

$$R(\hat{h}_\phi) \;\leq\; R(h^*) \;+\; 4\,\text{Rad}_m(\mathcal{L}_{meta}(\phi) \circ \mathcal{H}) \;+\; M\sqrt{\tfrac{2\ln(1/\delta)}{m}},$$

*where $R(h_\phi)$ and $\widehat{R}_m(h_\phi)$ denote the true and empirical risk respectively, and $\text{Rad}_m$ is the Rademacher complexity of the meta-loss hypothesis. The detailed proof is provided in Appendix A.2.*

Theorem 1 shows that as the meta-buffer size $m$ increases, the risk of the learned meta-learner $R(\hat{h}_\phi)$ converges to that of the oracle function $R(h^*)$. And the Rademacher complexity term favors a simple hypothesis space for better generalization. This justifies that a sufficient meta-buffer combined with a simple meta-learner can achieve effective estimation and assign near-optimal weights.

## 5 EXPERIMENTS

### 5.1 EXPERIMENTAL SETUP

**Datasets and Models.** We use the Ultrafeedback dataset (Cui et al., 2024) , which contains 64K preference annotations collected from four LLM-generated outputs and labeled preferences by GPT-4. To demonstrate the general effectiveness of our method, we experiment with two supervised fine-tuned models from different-sized base model: Llama-3.1-8B (Grattafiori et al., 2024) and

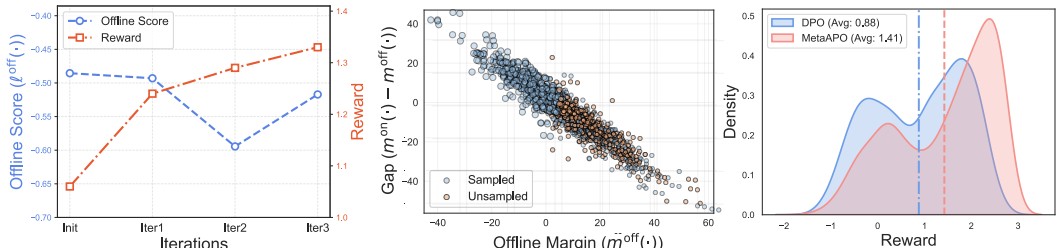

Figure 2: **Left:** The dynamic changes in Offline Score (left y-axis) and Reward (right y-axis) across training iterations. **Middle:** Scatter plot of offline implicit reward margin ($m_{off}(\cdot)$) versus the online-offline implicit reward margin gap ($m_{on}(\cdot) - m_{off}(\cdot)$). Points are colored by their sampling status: blue for *"Sampled"* (selected for online generation) and orange for *"Unsampled"*. **Right:** Comparison of independent reward score distributions (via kernel density estimation) for testset responses generated by DPO and MetaAPO.

Qwen2.5-7B (Yang et al., 2024). For online preference annotation, we train the reward models on UltraFeedback using the corresponding base model to avoid any preference leakage. We also provide a comparison with existing open-source reward models in Appendix C.2.

**Evaluation Metric.** We evaluate our method using three widely adopted automatic evaluation benchmarks for LLMs: AlpacaEval 2 (Dubois et al., 2024), MT-Bench (Zheng et al., 2023), and Arena-Hard (Li et al., 2024). All evaluations are conducted using GPT-4o-latest as the judge model. For AlpacaEval 2, we report the raw win rate (WR) and the length-controlled win rate (LC) against the reference model GPT-4o-2024-05-13. For Arena-Hard, we report the win rate (WR) and the style-controlled win rate (SC), comparing our model against the GPT-4-Preview-1106 baseline. For MT-Bench, we report the average multi-turn score (Score) assigned by GPT-4o-latest, which rates each response on a 10-point scale.

**Baselines.** We evaluate the following baseline methods relevant to the setting of our method:

- **Offline Alignment**: Methods that optimize models solely using offline human preference datasets, such as DPO (Rafailov et al., 2023), IPO (Azar et al., 2024), KTO (Ethayarajh et al., 2024) and DPO-Shift (Yang et al., 2025). This category also includes Selective DPO (Gao et al., 2025), which improves data quality by filtering out OOD samples based on a hand-crafted metric.
- **Online Alignment**: Methods that collect model-generated responses and align them using preferences from an external reward model—*e.g.,* Online DPO (Xiong et al., 2023) and PPO.
- **Hybrid Alignment**: Methods that integrate both offline annotated preferences and online model-generated data to optimize alignment, as ADPO (Ji et al., 2024) and SELM (Zhang et al., 2024b). It also includes methods that apply fixed-threshold data filtering before performing online augmentation—*e.g.,* BeeS (Deng et al., 2025) and MAP (Huang et al., 2025).

**Implementation.** All experiments are conducted using $8 \times 80$GB NVIDIA H100 GPUs with a batch size of 128 and a maximum sequence length of 4096 tokens for 1 epoch. At each iteration round, we sample responses from the current policy using different decoding configurations: temperature $T = 1.0$, $top\text{-}p = 1.0$ for the Llama3.1 setting, and $T = 0.8$, $top\text{-}p = 0.95$ for the Qwen2.5 setting. For each prompt, we generate $K = 8$ diverse responses, which are then scored by a reward model to construct preference pairs. The meta-learner is updated every $T_{meta} = 8$ steps. Additional details about model configurations, evaluation procedure and hyperparameters are provided in Appendix B.

## 5.2 OVERALL PERFORMANCE

In Table 1, we present the overall results of MetaAPO compared to offline, online and hybrid state-of-the-art alignment methods on the AlpacaEval 2, Arena-Hard and MT-Bench benchmarks.

**MetaAPO consistently outperforms preference optimization baselines.** Compared to offline alignment methods, which rely solely on fixed, pre-collected datasets, MetaAPO avoids the performance degradation caused by distribution shifts between static data and evolving policies. Enhanced offline approaches like Selective DPO, which attempt to mitigate this mismatch through data filtering, yield

Table 1: Overall performance of our proposed MetaAPO method with Llama-3.1-8B and Qwen2.5-7B, compared with offline, online and hybrid baseline methods on AlpacaEval 2, Arena-Hard and MT-Bench. The best results are highlighted in **bold**.

| Models | Llama-3.1-8B | | | | | Qwen2.5-7B | | | | |
|---|---|---|---|---|---|---|---|---|---|---|
| | AlpacaEval 2 | | Arena-Hard | | MT-Bench | AlpacaEval 2 | | Arena-Hard | | MT-Bench |
| | WR(%) | LC(%) | WR(%) | SC(%) | Score | WR(%) | LC(%) | WR(%) | SC(%) | Score |
| **SFT** | 10.02 | 17.28 | 18.5 | 21.6 | 6.63 | 34.03 | 34.08 | 48.5 | 44.3 | 7.71 |
| *Offline Alignment* | | | | | | | | | | |
| **DPO** | 18.15 | 25.20 | 28.9 | 30.9 | 6.94 | 37.24 | 36.84 | 49.0 | 47.2 | **7.83** |
| **IPO** | 13.11 | 23.50 | 23.6 | 26.3 | 7.02 | 37.95 | 36.43 | 54.6 | 48.3 | 7.64 |
| **KTO** | 20.28 | 29.19 | 21.4 | 26.4 | 7.24 | 38.12 | 36.51 | 43.9 | 44.1 | 7.63 |
| **SimPO** | 37.31 | 38.20 | 37.8 | 33.0 | 7.51 | 40.03 | 40.78 | 54.6 | 48.8 | 7.58 |
| **DPO-Shift** | 19.33 | 19.39 | 28.7 | 27.6 | 7.03 | 36.05 | 34.32 | 49.6 | 47.2 | 7.53 |
| **Selective DPO** | 21.35 | 20.67 | 28.5 | 31.9 | 7.11 | 38.02 | 39.21 | 51.7 | 48.2 | 7.74 |
| *Online & Hybrid Alignment* | | | | | | | | | | |
| **ADPO** | 19.33 | 17.26 | 32.0 | 30.7 | 6.69 | 40.32 | 35.21 | 49.9 | 48.7 | 7.58 |
| **Online DPO** | 43.75 | 39.98 | 38.0 | 32.1 | 7.33 | 48.25 | 46.64 | 53.7 | 48.4 | 7.51 |
| **PPO** | 45.33 | 40.49 | 40.8 | 35.3 | 7.35 | 48.52 | 44.26 | 55.1 | 49.2 | 7.65 |
| **SELM** | 43.56 | 40.14 | 36.0 | 34.1 | 7.28 | 48.08 | 44.91 | 56.5 | 51.7 | 7.48 |
| **BeeS** | 41.42 | 37.45 | 39.5 | 36.4 | 7.13 | 41.87 | 37.04 | 49.3 | 48.2 | 7.38 |
| **MAP** | 43.60 | 38.97 | 40.4 | 37.2 | 7.17 | 42.11 | 39.91 | 52.1 | 49.5 | 7.52 |
| *MetaAPO* | | | | | | | | | | |
| **MetaAPO** | **47.48** | **43.21** | **43.9** | **40.8** | **7.56** | **49.14** | **47.66** | **58.4** | **53.7** | 7.78 |

only marginal improvements and still fall short of MetaAPO. Online and hybrid methods leverage model-generated samples and online feedback to significantly outperform purely offline baselines. While online and hybrid methods significantly outperform purely offline baselines by leveraging on-policy feedback, they are often constrained by the quality and diversity of their own model generations. The heavy reliance on self-generated data can introduce noisy preferences. MetaAPO surpasses these strong baselines by avoiding regions prone to noisy feedback and effectively leveraging diverse offline data. We also compare performance across different reward models, preference optimization methods and datasets in Appendix C.2, demonstrating the robustness of our approach.

**MetaAPO achieves superior performance with fewer online samples.** As shown in Figure 1 (Right), MetaAPO outperforms strong baselines such as Online DPO, SELM and PPO, while using only 58% of the online generated and annotated samples on average compared to standard online generation methods. This efficiency comes from a meta-weighted sampling mechanism, which prioritizes the most informative data points and skips unnecessary online generation for already aligned or low-gain predicted samples. This adaptive data sampling significantly reduces labeling costs and time consumption, cutting time by 80.1% compared to PPO and by 52.9% compared to Online DPO (detailed in Appendix C.7).

## 5.3 FURTHER ANALYSES

**Training dynamics reveal an explore-integrate behavior.** To examine the dynamic interplay during training, we tracked the offline preference score $\ell^{\text{off}}$ and generation reward on a held-out test set across iterations. As shown in Figure 2 (Left), the average reward steadily increases (from 1.06 to 1.33) while its standard deviation decreases (from 0.49 to 0.37), indicating that MetaAPO robustly enhances generation quality under reward guidance. In contrast, the mean offline preference score drops sharply at Iteration 2 before recovering, suggesting that MetaAPO does more than fit static offline data. The drop reflects active exploration of response strategies diverging from offline preferences, while the subsequent recovery in Iteration 3 reflects the integration of these discoveries with foundational offline knowledge. Overall, these dynamics highlight MetaAPO's adaptive behavior: it continuously identifies and addresses gaps between the evolving policy and offline data.

Table 2: Ablation study on key components of MetaAPO. We assess the impact of the sampling strategy, loss weighting, and meta-learner. Metrics include AlpacaEval2, Arena-Hard, MT-Bench and the proportion of annotations used during training.

| Model | AlpacaEval 2 | | Arena-Hard | | MT-Bench | Annotations |
| | WR(%) ↑ | LC(%) ↑ | WR(%) ↑ | SC(%) ↑ | Score ↑ | Ratio(%) ↓ |
|---|---|---|---|---|---|---|
| **MetaAPO** | **47.48** | **43.21** | **43.9** | **40.8** | **7.56** | 58.4 |
| *w/* Random Sampling | 42.61 | 38.53 | 37.3 | 32.1 | 7.14 | 50.0 |
| *w/* Threshold Sampling | 44.13 | 42.92 | 40.7 | 38.5 | 7.14 | 53.3 |
| *w/* All Sampling | 46.13 | 42.04 | 40.2 | 37.5 | 6.96 | 100.0 |
| *w/* Uniform Loss Weighting | 39.25 | 37.70 | 39.2 | 33.3 | 7.15 | 67.1 |
| *w/* Semantic Similarity | 47.83 | 42.88 | 43.7 | 40.5 | 7.50 | 59.4 |
| *w/* Fixed Meta-Learner | 45.01 | 40.03 | 39.9 | 36.4 | 7.22 | 60.6 |
| *w/o* Meta-Learner | 43.07 | 40.48 | 38.3 | 33.1 | 7.06 | 61.1 |

**The dynamics are driven by a high-gain sampling strategy.** Figure 2 (Middle) shows MetaAPO's online sampling strategy based on two signals: the initial offline preference score $\ell^{\text{off}}(\cdot)$, and the potential gain $\ell^{\text{on}}(\cdot) - \ell^{\text{off}}(\cdot)$. For visualization purposes, we replace the original preference score $\ell(\cdot) = \log \sigma(\beta \cdot m(\cdot))$ with the implicit reward margin $m(\cdot) = \log \frac{\pi_{\boldsymbol{\theta}}(\boldsymbol{y}_w|\boldsymbol{x})}{\pi_{\text{ref}}(\boldsymbol{y}_w|\boldsymbol{x})} - \log \frac{\pi_{\boldsymbol{\theta}}(\boldsymbol{y}_l|\boldsymbol{x})}{\pi_{\text{ref}}(\boldsymbol{y}_l|\boldsymbol{x})}$, which allows us to more clearly visualize the distribution in well-aligned regions (right x-axis). This rescaling only changes the numerical magnitude of the values and does not affect their relative ordering or the qualitative sampling trends. Each point represents an offline sample, marked as *"Sampled"* (blue) if selected for generation, or *"Unsampled"* (orange) otherwise. Two trends drive sampling. First, points with low $m^{\text{off}}(\cdot)$ (indicating weak alignment or OOD) receive smaller meta-learner weights $w$, increasing their selection probability. Second, the meta-learner prioritizes samples with high potential gain ($m^{\text{on}}(\cdot) > m^{\text{off}}(\cdot)$), concentrating its selection in the upper-left region where these two factors converge. In contrast, samples with high offline scores or minimal online gain are less likely to be selected, avoiding redundant computation.

**Effectiveness of the meta-learner.** A key component of MetaAPO is the meta-learner, a simple two-layer MLP that effectively captures the model's dynamic alignment state. Its effectiveness stems from two factors. First, we observed a clear, learnable relationship between the offline score $\ell^{\text{off}}(\cdot)$ and potential online-offline gain $\ell^{\text{on}}(\cdot) - \ell^{\text{off}}(\cdot)$, as shown in Figure 2 (Middle). This discovery allows even a lightweight network to serve as a reliable "alignment gap estimator." Second, the meta-learner is continuously updated alongside the policy, adapting its sampling and weighting decisions to the evolving model state. Experiments with deeper networks and more complex inputs (Appendix C.5) show that the two-layer MLP provides sufficient expressive power, confirming its efficiency and effectiveness. Additional analyses further confirm the generalizability of MetaAPO. As shown in Figure 2 (Right), its generated responses on the test set achieve higher scores distribution under an independent reward model OpenRLHF/Llama-3-8b-rm-mixture.

## 5.4 ABLATION STUDY

To analyze the impact of each core component in MetaAPO, we conduct a series of ablation studies under the Llama-3.1-8B setting with the DPO objective. Table 2 summarizes the effectiveness of different sampling strategies, loss weighting mechanisms and the meta-learner.

**Effects of sampling strategies.** The choice of which offline samples to augment with online responses is a critical factor to achieve efficient and effective alignment. We ablate three design choices and keep the preference optimization and meta-learner training process consistent with the original MetaAPO setup: (1) *Random Sampling* selects half of the offline data uniformly for online augmentation. (2) *Threshold Sampling* selects samples whose initial preference scores fall below a fixed threshold (resulting in a 53.3% annotation ratio). (3) *All Sampling* disables the meta-learner's adaptive sampling mechanism and applies online sampling to all samples without any selection. The meta loss weighting and meta-learner update still follow the MetaAPO formulation. As shown in Table 2, MetaAPO

outperforms both random and threshold sampling across all metrics. *All Sampling* leverages a larger volume of data, yet the absence of consideration for the dynamic distribution relationship between offline and online data during sampling results in suboptimal alignment performance.

**Effects of loss weighting.** We evaluate the impact of meta-weighted preference optimization by replacing the learned weights $w$ in Eq. 5 with a *uniform coefficient* of $0.5$ for all augmented samples, assigning equal importance to offline and online preference pairs during training. Other components, including adaptive sampling and meta-learner updates, remain unchanged. As shown in Table 2, this modification leads to a significant performance drop with consistent declines on all benchmarks. These results suggest that assigning equal importance forces the model to learn from low-margin online data for already-aligned samples, which provides a weak signal and can destabilize learning, underscoring the need for a dynamic balancing mechanism.

**Effects of meta-learner.** To investigate the informative and adaptive design of the meta-learner, we ablate three variants while keeping the overall preference optimization and training pipeline consistent with MetaAPO: (1) *Semantic Similarity* augments the meta-learner input by incorporating the embedding similarity between the chosen and rejected responses from the offline data. Results show that adding semantic similarity leads to comparable performance, indicating that the original preference score already provides sufficient information for meta-learner. (2) *Fixed Meta-Learner* initialize a new meta-learner with the weights learned by MetaAPO and freeze it during training. Both sampling weights and loss weights remain static. (3) We also evaluate the necessity of a trainable meta-learner by replacing meta-learner function $h_\phi$ with a fixed sigmoid-based heuristic (*w/o Meta-Learner* in Table 2). This heuristic also uses offline preference scores as input but remains static during the meta-learner update phase. The results show performance drops across all metrics. For example, AlpacaEval 2 WR decreased from $47.48\%$ to $43.07\%$, and Arena-Hard WR dropped from $43.9\%$ to $38.3\%$. Despite using more online annotations ($61.1\%$ *vs.* $58.4\%$), the fixed heuristic underperformed, confirming the importance of a learnable meta-learner for adaptive weighting and effective alignment. Further details, along with additional ablation studies on reward model annotations and offline data usage, are provided in Appendix C.2 and C.5.

## 6 CONCLUSION

In this paper, we propose MetaAPO, a novel framework that bridges the gap between data generation and model training in LLM alignment. The core idea is to leverage a meta-learner to assess the alignment of the current policy and identify the potential for online exploration. To achieve this, we introduce the techniques of *adaptive online sampling*, *meta-weighted optimization* and *meta-learner update*, which jointly enable dynamic integration of data and training. Experiments across three alignment benchmarks validate the effectiveness of MetaAPO in consistently outperforming existing alignment methods. While our current approach effectively utilizes preference scores as input, it may be beneficial to explore additional input signals, such as gradient or representation-based features, to further enhance flexibility and generalization.

### ACKNOWLEDGMENTS

This research was supported by the Jiangsu Science Foundation (BG2024036, BK20243012), the National Science Foundation of China (62576093, 62125602, U24A20324, and 92464301), the Fundamental Research Funds for the Central Universities (2242025K3002, 2242025K30024), the New Cornerstone Science Foundation through the XPLORER PRIZE, and the Big Data Computing Center of Southeast University.

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

APPENDIX

In the appendix, we provide the following supplementary materials: (A) Details of the formulation derivation, including meta-learner gradient derivation, theorem proof and SimPO objective formulation. (B) Implementation details of the experimental setup, including models and datasets, evaluation benchmarks, baselines, model training and hyperparameters. (C) presents further experimental analyses, including robustness of reward models, our tuning strategy for the meta-learner update interval $T_{\text{meta}}$, deeper investigations into the meta-learner's behavior with supporting visualizations, and results from an extended ablation study. (D) provides qualitative case studies that illustrate MetaAPO's output quality in comparison to baseline methods.

## A  MATHEMATICAL DERIVATIONS AND FORMULATIONS

### A.1  DERIVATION OF THE META-LEARNER LOSS GRADIENT

The meta-learner aims to capture the trade-off between offline and online preference and maximize the expected weighted sum of these preference scores. Therefore, its update is guided by minimizing the following loss:

$$\mathcal{L}_{\text{meta}}(\phi) = -\mathbb{E}_{\mathcal{B}_{\text{meta}}}\Big[h_\phi(\ell^{\text{off}}(\cdot)) \cdot \ell^{\text{off}}(\cdot) + (1 - h_\phi(\ell^{\text{off}}(\cdot))) \cdot \ell^{\text{on}}(\cdot)\Big]. \tag{A.1}$$

This expression can be reformulated to more effectively represent the optimal objective $h_\phi(\ell^{\text{off}}(\cdot))$:

$$\begin{aligned}
\mathcal{L}_{\text{meta}}(\phi) &= -\mathbb{E}_{\mathcal{B}_{\text{meta}}}\Big[\big(\ell^{\text{off}}(\cdot) - \ell^{\text{on}}(\cdot)\big) \cdot h_\phi(\ell^{\text{off}}(\cdot)) + \ell^{\text{on}}(\cdot)\Big] \\
&= \mathbb{E}_{\mathcal{B}_{\text{meta}}}\Big[\big(\ell^{\text{on}}(\cdot) - \ell^{\text{off}}(\cdot)\big) \cdot h_\phi(\ell^{\text{off}}(\cdot)) - \ell^{\text{on}}(\cdot)\Big].
\end{aligned} \tag{A.2}$$

Differentiating both sides of $\mathcal{L}_{\text{meta}}$ with respect to the meta-learner parameters $\phi$ yields:

$$\nabla_\phi \mathcal{L}_{\text{meta}}(\phi) = \mathbb{E}_{\mathcal{B}_{\text{meta}}}\Big[\big(\ell^{\text{on}}(\cdot) - \ell^{\text{off}}(\cdot)\big) \cdot \nabla_\phi(h_\phi(\ell^{\text{off}}(\cdot)))\Big]. \tag{A.3}$$

This gradient governs the meta-learner's update. The term $(\ell^{\text{on}}(\cdot) - \ell^{\text{off}}(\cdot))$ can be seen as an "advantage" signal, indicating how much better (or worse) the online score is compared to the offline score for the same underlying prompt.

**Interpretation.** When online samples yield higher preference scores than their offline counterparts (*i.e.*, $\ell^{\text{on}}(\cdot) > \ell^{\text{off}}(\cdot)$), the factor $(\ell^{\text{on}}(\cdot) - \ell^{\text{off}}(\cdot))$ is positive. The gradient update will then tend to adjust $\phi$ such that $h_\phi(\ell^{\text{off}}(\cdot))$ (the weight assigned to the offline sample) decreases. This encourages more reliance on online exploration and data for instances where it proves more beneficial. Conversely, when offline samples yield scores that are comparable to or better than online samples (*i.e.*, $\ell^{\text{off}}(\cdot) \geq \ell^{\text{on}}(\cdot)$), the factor $\ell^{\text{on}}(\cdot) - \ell^{\text{off}}(\cdot)$ is zero or negative. This leads to updates that preserve or increase the weight $h_\phi(\ell^{\text{off}}(\cdot))$, thereby promoting stable exploitation of reliable, curated offline data.

### A.2  PROOF OF THEOREM 1

We provide a complete proof of Theorem 1. First, we establish a uniform convergence bound between the empirical and true risk of the meta-learner, using the Rademacher complexity of the meta-loss hypothesis class. Then, we decompose the excess risk of the learned meta-learner relative to the oracle into terms that can be bounded by the uniform convergence result. Finally, we obtain a high-probability generalization guarantee for $\hat{h}_\phi$ as stated in the theorem.

### A.2.1  PRELIMINARIES

**Definition 1** (Meta-Loss Function). *For a meta-sample $(\boldsymbol{x}, \boldsymbol{y}_{pair}^{off}, \boldsymbol{y}_{pair}^{on})$ and a meta-learner $h_\phi$, the meta-loss is defined as:*

$$\mathcal{L}_{meta}(h_\phi; \boldsymbol{x}, \boldsymbol{y}_{pair}^{off}, \boldsymbol{y}_{pair}^{on}) \in [0, M].$$

*where $M$ is an upper bound on the loss. The meta-loss is always non-negative and does not exceed $M$ for any samples $(\boldsymbol{x}, \boldsymbol{y}_{pair}^{off}, \boldsymbol{y}_{pair}^{on})$.*

**Definition 2** (True Risk and Empirical Risk). *Let $\mathcal{D}$ be the distribution over meta-samples, and $\mathcal{B}_{meta} = \{(\boldsymbol{x}_i, \boldsymbol{y}_i^{off}, \boldsymbol{y}_i^{on})\}_{i=1}^m$ a meta-buffer of size $m$. The true and empirical risk of $h_\phi$ are defined as:*

$$R(h_\phi) = \mathbb{E}_{(\boldsymbol{x}, \boldsymbol{y}_{pair}^{off}, \boldsymbol{y}_{pair}^{on}) \sim \mathcal{D}}[\mathcal{L}_{meta}(h_\phi)], \quad \widehat{R}_m(h_\phi) = \frac{1}{m} \sum_{i=1}^m \mathcal{L}_{meta}(h_\phi; \boldsymbol{x}_i, \boldsymbol{y}_i^{off}, \boldsymbol{y}_i^{on}).$$

**Definition 3** (Rademacher Complexity). *For a function class $\mathcal{F}$, the empirical Rademacher complexity with respect to the meta-buffer $\mathcal{B}_{meta}$ is:*

$$\widehat{\mathrm{Rad}}_m(\mathcal{F}) = \mathbb{E}_\sigma \Big[ \sup_{f \in \mathcal{F}} \frac{1}{m} \sum_{i=1}^m \sigma_i f(\boldsymbol{x}_i, \boldsymbol{y}_i^{off}, \boldsymbol{y}_i^{on}) \Big| \mathcal{B}_{meta} \Big],$$

*where $\sigma_i$ are i.i.d. Rademacher variables. Its expectation over $\mathcal{B}_{meta}$ defines $\mathrm{Rad}_m(\mathcal{F})$. This quantity captures the capacity/complexity of the meta-loss hypothesis class and is a standard tool in statistical learning theory (Mohri et al., 2018).*

### A.2.2 PROOF OF THEOREM 1

**Step 1: Uniform Convergence.** By standard results in statistical learning theory (Mohri et al., 2018), for any function class $\mathcal{F}$ bounded in $[0, M]$, with probability at least $1 - \delta$ over the choice of the meta-buffer $\mathcal{B}_{\mathrm{meta}}$:

$$\sup_{f \in \mathcal{F}} \left| \mathbb{E}[f] - \widehat{\mathbb{E}}_m[f] \right| \leq 2 \mathrm{Rad}_m(\mathcal{F}) + M \sqrt{\tfrac{\ln(1/\delta)}{2m}}. \tag{A.4}$$

**Step 2: Excess Risk Decomposition.** Let $\Delta = \sup_{h_\phi \in \mathcal{H}} |R(h_\phi) - \widehat{R}_m(h_\phi)|$. For the empirical minimizer $\hat{h}_\phi = \arg\min_{h_\phi \in \mathcal{H}} \widehat{R}_m(h_\phi)$ and the oracle $h^* = \arg\min_{h_\phi \in \mathcal{H}} R(h_\phi)$, we have

$$
\begin{aligned}
R(\hat{h}_\phi) - R(h^*) &= \underbrace{R(\hat{h}_\phi) - \widehat{R}_m(\hat{h}_\phi)}_{\leq \Delta} + \underbrace{\widehat{R}_m(\hat{h}_\phi) - \widehat{R}_m(h^*)}_{\leq 0} + \underbrace{\widehat{R}_m(h^*) - R(h^*)}_{\leq \Delta} \\
&\leq 2 \sup_{h_\phi \in \mathcal{H}} |R(h_\phi) - \widehat{R}_m(h_\phi)|.
\end{aligned}
\tag{A.5}
$$

**Step 3: Substituting the Uniform Bound.** Applying (A.4) to the meta-loss class $\mathcal{L}_{\mathrm{meta}}(\phi) \circ \mathcal{H}$, with probability at least $1 - \delta$:

$$\sup_{h_\phi \in \mathcal{H}} |R(h_\phi) - \widehat{R}_m(h_\phi)| \leq 2 \mathrm{Rad}_m(\mathcal{L}_{\mathrm{meta}}(\phi) \circ \mathcal{H}) + M \sqrt{\frac{\ln(1/\delta)}{2m}}. \tag{A.6}$$

Combining with (A.5) yields the desired bound:

$$R(\hat{h}_\phi) \leq R(h^*) + 4 \mathrm{Rad}_m(\mathcal{L}_{\mathrm{meta}}(\phi) \circ \mathcal{H}) + M \sqrt{\frac{2 \ln(1/\delta)}{m}}. \tag{A.7}$$

This completes the proof. $\qquad\square$

### A.3 METAAPO WITH SIMPO OBJECTIVE

The MetaAPO framework is designed to be largely agnostic to the choice of the underlying preference optimization loss, enabling it to be applied in conjunction with various alignment algorithms. To showcase this flexibility, we demonstrate its integration with Simple Preference Optimization (SimPO) (Meng et al., 2024), a reference-free alignment method. This integration results in the MetaAPO-SimPO variant.

The original objective of SimPO involves minimizing the following loss function:

$$\mathcal{L}_{\mathrm{SimPO}}(\pi_{\boldsymbol{\theta}}) = -\mathbb{E}_{(\boldsymbol{x}, \boldsymbol{y}_w, \boldsymbol{y}_l) \sim \mathcal{D}} \left[ \log \sigma \left( \frac{\beta}{|\boldsymbol{y}_w|} \log \pi_{\boldsymbol{\theta}}(\boldsymbol{y}_w|\boldsymbol{x}) - \frac{\beta}{|\boldsymbol{y}_l|} \log \pi_{\boldsymbol{\theta}}(\boldsymbol{y}_l|\boldsymbol{x}) - \gamma \right) \right]. \tag{A.8}$$

From this formulation, the corresponding instance-level preference score $\ell(\boldsymbol{x}, \boldsymbol{y}_w^{\text{off}}, \boldsymbol{y}_l^{\text{off}})$ in SimPO, used to quantify how well the current policy $\pi_{\boldsymbol{\theta}}$ aligns with the given preference pair $(\boldsymbol{y}_w, \boldsymbol{y}_l)$ for a prompt $\boldsymbol{x}$, is defined as the term whose negative log-likelihood is minimized in the SimPO objective:

$$\ell(\boldsymbol{x}, \boldsymbol{y}_w, \boldsymbol{y}_l) = \log \sigma \left( \frac{\beta}{|\boldsymbol{y}_w|} \log \pi_{\boldsymbol{\theta}}(\boldsymbol{y}_w|\boldsymbol{x}) - \frac{\beta}{|\boldsymbol{y}_l|} \log \pi_{\boldsymbol{\theta}}(\boldsymbol{y}_l|\boldsymbol{x}) - \gamma \right). \tag{A.9}$$

When applying MetaAPO in conjunction with SimPO (*i.e.,* MetaAPO-SimPO), this SimPO-specific preference score (as shown in Eq. A.9) is used in place of the DPO-based preference score within the MetaAPO framework.

## B  EXPERIMENTAL SETUP

### B.1  MODELS AND DATASET

For the SFT models used in training, we adopt existing open-source models that have been supervised fine-tuned, specifically: allenai/Llama-3.1-Tulu-3-8B-SFT based on the Llama architecture, and FuseAI/FuseChat-Qwen-2.5-7B-SFT based on the Qwen architecture. All methods are trained on UltraFeedback-Binarized dataset, which contains 64K preference data.

**Reward models.** For the external reward model annotation, we trained the reward model for one epoch on the UltraFeedback-Binarized dataset from the SFT model corresponding to the policy model. This procedure ensures that any online improvement does not stem from potential leakage of the reward model, thus providing a fully fair comparison between online and offline methods. We also report experimental results using other open-source reward models in the Appendix.

**Meta-Learner architecture.** The meta-learner $h_{\boldsymbol{\phi}}$ is a compact two-layer MLP that maps the scalar offline preference score $\ell^{\text{off}}(\cdot)$ to a weight $w \in [0, 1]$. It uses a 100-dimensional hidden layer with Tanh activation, followed by a Sigmoid output. Due to its small size and simple input, the meta-learner adds negligible overhead in both prediction and training.

### B.2  EVALUATION BENCHMARKS

We evaluate our models on three representative alignment benchmarks: AlpacaEval 2(Dubois et al., 2024), MT-Bench(Zheng et al., 2023), and Arena-Hard (Li et al., 2024). These benchmarks are widely adopted for their comprehensive coverage of diverse tasks and their effectiveness in assessing the subjective preference capabilities of LLMs.

- **AlpacaEval 2** includes 805 instructions from five different datasets and assesses models using two metrics: length-controlled (LC) win rate and raw win rate (WR). In this benchmark, GPT-4o-2024-05-13 serves as the baseline model and GPT-4o-latest as the evaluator for the other models.

- **Arena-Hard** is a more challenging benchmark that closely aligns with the human preference ranking from Chatbot Arena, a crowd-sourced platform for evaluating LLMs. It spans 250 high-quality topic clusters, including 500 well-defined technical problem-solving queries. We report the Win Rate against GPT-4-0314 using GPT-4o as the judge model.

- **MT-Bench** contains 80 multi-turn dialogues with 160 questions across eight categories, including writing, roleplay, reasoning, math, coding, extraction, STEM, and humanities. Each response is evaluated by GPT-4o on a scale from 1 to 10, with the average score reported for each dialogue turn across the 80 dialogues. Different from the official setting, we follow the latest works to adopt GPT-4o as the evaluator and baseline.

### B.3  BASELINES.

To evaluate the effectiveness of MetaAPO, we compare it against a range of SOTA baseline methods from current LLM alignment literature. These baselines include purely offline approaches, methods that enhance offline data processing, techniques incorporating online data generation, and hybrid strategies:

- **Offline Preference Alignment**: These methods, such as DPO (Rafailov et al., 2023), SimPO (Meng et al., 2024), and IPO (Azar et al., 2024), directly learn from a fixed, pre-collected preference dataset, typically by optimizing a log-likelihood objective.

- **Selective DPO** (Gao et al., 2025): This approach aims to improve data quality by splitting the dataset into training and validation sets. It calculates the average loss of all samples on the validation set and then filters out samples presumed to be low-quality (those with higher validation loss) to curate a sub-dataset for model training.

- **Online DPO** (Xu et al., 2024): Online DPO follows an iterative process of data generation and preference optimization. In each iteration, the current model generates multiple responses for given prompts. These responses are subsequently evaluated by an external reward model to identify the best (chosen, $y_w^{\text{on}}$) and worst (rejected, $y_l^{\text{on}}$) options, forming new preference pairs. The policy optimization then exclusively utilizes these self-generated preference pairs. Then the policy model and reference model are updated and continue next iteration.

- **ADPO** (Ji et al., 2024): Active DPO (ADPO) heuristically selects offline samples for online augmentation if their preference loss falls below a fixed threshold. New online preference pairs are generated for these selected prompts. These newly created online pairs are then used in conjunction with the original offline samples for preference optimization via a standard DPO algorithm.

- **SELM** (Zhang et al., 2024b): Self-Exploring Language Model (SELM) operates with an iterative generation and annotation cycle similar to Online DPO. However, when ranking the multiple responses generated by the current policy (using a reward model), SELM distinctively includes the original chosen $y_w^{\text{off}}$ and rejected $y_l^{\text{on}}$ responses from the offline dataset (for the same prompt) within the ranking pool. This method allows for a more integrated utilization of both offline and newly generated online data to construct the training preference pairs.

- **PPO** (Ouyang et al., 2022): Proximal Policy Optimization (PPO) is a foundational online alignment method from Reinforcement Learning from Human Feedback. Unlike direct preference methods, PPO first trains a separate reward model on a human preference dataset. Subsequently, the policy model generates responses to prompts, and the RM scores these responses to provide a reward signal. PPO then uses this reward to update the policy and critic model, aiming to maximize the expected reward while a KL-divergence penalty keeps the policy from straying too far from a reference model.

- **DPO-Shift** (Yang et al., 2025): DPO-Shift is an enhanced direct preference optimization approach designed to address the "likelihood shift" issue inherent in traditional DPO methods. By introducing a parametric function $f(\lambda)$ into the Bradley-Terry model, DPO-Shift enables controlled adjustment of the probability distribution for selected responses. Specifically, it adds $f(\lambda)$ to the reward of rejected responses, thereby reducing the simultaneous decline in probabilities of both preferred and non-preferred responses while maximizing the reward gap between them. In our experiments, we used the parameters that reported the best performance in the paper, with $f(\lambda)$ linearly increasing from $0.95$ to $1.0$.

- **BeeS** (Deng et al., 2025): BeeS is an enhanced data selection strategy designed to address OOD issue inherent in DPO training caused by noisy preference data. By introducing a Bayesian aggregation approach into the margin-based selection framework, BeeS enables robust identification of high-quality training samples. Specifically, it unifies multiple reward margin sources, including both external reward models and implicit DPO rewards, prioritizing preference pairs that exhibit low reward margins from any single source to mitigate potential noise. In our experiments, we used the 25% sample ratio that demonstrated optimal efficiency in the paper.

- **MAP** (Huang et al., 2025): MAP is an enhanced data selection metric designed to address the contradictory evaluation signals inherent in preference data selection. By introducing a gap-based quantification approach into the margin-based selection framework, MAP enables the identification of high-quality training samples. Specifically, it unifies explicit reward model margins and implicit policy reward margins, prioritizing preference pairs where the target explicit reward substantially exceeds the current implicit reward margin, thereby precisely estimating alignment enhancement potential. In our experiments, we used the 40% sample ratio that demonstrated optimal performance as reported in the paper.

Table 3: Hyperparameters for Llama-3.1-8B and Qwen2.5-7B during generation and training.

| Hyperparameters | Notation | Llama-3.1-8B | Qwen2.5-7B |
|---|---|---|---|
| *Generation* | | | |
| Temperature | - | 1 | 0.8 |
| Top-p | - | 1 | 0.95 |
| Rollout nums | K | 8 | |
| Max new token | $L_{\text{new}}$ | 2048 | |
| *Training* | | | |
| Learning rate | $\alpha$ | 5e-7 | |
| Meta-learner lr | $\eta$ | 5e-3 | |
| Batch size | B | 128 | |
| Max prompt length | $L_{\text{prompt}}$ | 2048 | |
| Max generation length | $L_{\text{gen}}$ | 2048 | |
| Training max length | $L_{\text{train}}$ | 4096 | |
| Reward model max length | $L_{\text{reward}}$ | 4096 | |
| KL loss | $\beta$ | 0.1 (2.5 for SimPO) | |
| Target reward margin | $\gamma$ | 0.6 | |
| Update Interval | $T_{\text{meta}}$ | 8 | |
| Epochs | - | 1 | |

## B.4 MODEL TRAINING

In our main experiments, we integrate MetaAPO into the iterative alignment framework (Ouyang et al., 2022; Xu et al., 2024; Dong et al., 2024; Peng et al., 2025), which consists of two stages: data generation and preference optimization. To obtain the policy and reference models, we first warm up the SFT model on the UltraFeedback-Binarized dataset using the DPO loss. The warmed-up model serves as the initial policy model and the SFT model is used as the reference model for MetaAPO training. For a fair comparison, all models are trained for **1 epoch** over the dataset and run for 3 alignment iterations.

During each generation phase, we first employ the meta-learner to assign sampling probabilities for the offline data. Based on these probabilities, a subset of offline preference data is sampled. For each sampled prompt, the current policy model generates $K = 8$ candidate responses. An external reward model is then used to annotate scores for each prompt–response pair, from which we select the highest and lowest scoring responses as the chosen and rejected samples, respectively.

In the optimization phase, we jointly leverage both the original offline preference pairs and the newly generated online preference pairs for each augmented instance, forming the dataset $\mathcal{D}_{\text{aug}}$. These pairs are integrated through our meta-weighting mechanism, as defined in the preference optimization objective (Eq. 5). Specifically, for each instance $(\boldsymbol{x}, \boldsymbol{y}_{\text{pair}}^{\text{off}}, \boldsymbol{y}_{\text{pair}}^{\text{on}}) \in \mathcal{D}_{\text{aug}}$, the meta-learner computes a dynamic weight $w = h_{\boldsymbol{\phi}}(\ell_{\text{off}}(\boldsymbol{x}, \boldsymbol{y}_w^{\text{off}}, \boldsymbol{y}_l^{\text{off}}))$, which determines the relative importance of the offline component $\ell_{\boldsymbol{\theta}}(\boldsymbol{x}, \boldsymbol{y}_w^{\text{off}}, \boldsymbol{y}_l^{\text{off}})$ and the online component $\ell_{\boldsymbol{\theta}}(\boldsymbol{x}, \boldsymbol{y}_w^{\text{on}}, \boldsymbol{y}_l^{\text{on}})$ in the total loss.

The policy model $\pi_{\boldsymbol{\theta}}$ is then updated for one epoch over the weighted dataset $\mathcal{D}_{\text{aug}}$. Concurrently, the meta-learner $h_{\boldsymbol{\phi}}$ is updated every $T_{\text{meta}}$ steps using its dedicated meta-objective (Eq. 6), based on the data stored in the meta buffer $\mathcal{B}_{\text{meta}}$, as described in Algorithm 1.

## B.5 HYPERPARAMETERS

We list the key hyperparameters used in our experiments for both Llama-3.1-8B and Qwen2.5-7B in Table 3. The generation settings (*e.g.,* temperature, top-p, and number of responses per prompt) are adjusted to suit each model's characteristics, while training-related parameters such as learning

Table 4: We evaluated the performance of the Llama-3.1-8B aligned models on tasks related to mathematics, logical reasoning, general knowledge, *et al*. The numbers in parentheses indicate the few-shot examples given for each task, and the results are presented in the table.

| Models | GSM8K(8) | MMLU(5) | ARC(25) | EQBench(0) | TruthfulQA(0) | IFEval(0) | Average |
|---|---|---|---|---|---|---|---|
| SFT | 72.71 | 64.81 | 57.59 | 65.51 | 46.85 | **70.98** | 63.08 |
| DPO | 74.22 | 64.98 | 59.39 | **67.58** | 48.62 | 70.24 | 64.17 |
| Online DPO | 76.50 | 64.93 | 60.24 | 63.27 | 54.40 | 63.40 | 63.79 |
| SELM | 76.33 | 65.03 | 61.01 | 63.92 | 55.21 | 61.74 | 63.87 |
| MetaAPO | **77.18** | **65.15** | **61.43** | 67.27 | **55.47** | 68.21 | **65.79** |

rates, batch size, and sequence length are kept consistent to ensure fair comparison. For MetaAPO, we adopt a moderate meta-learner update interval and use shared weighting hyperparameters across all models and baselines.

For PPO training, we set the rollout batch size to 1024 to ensure a small distribution shift. All other hyperparameters, such as the batch size and learning rate, are kept consistent with those used in the other methods to enable a fair comparison.

**Evaluation.** We follow the generation protocols specified by each benchmark. The maximum generation length is set to 4096, while the temperature is determined according to the default configuration of the respective benchmarks.

## C ADDITIONAL EXPERIMENTAL RESULTS AND ANALYSES

In this section, we provide extended experiments and analyses to further validate MetaAPO. (C.1) examines its impact on general downstream tasks to ensure no degradation of broad capabilities. (C.2) test robustness under different reward models and datasets. (C.4) analyzes the effect of the meta-learner update interval and its dynamic behavior during training. Finally, we conduct extended ablation studies (C.5) and report time efficiency comparisons (C.7).

### C.1 PERFORMANCE ON GENERAL TASKS

We evaluated the aligned model's performance on other tasks involving mathematics, logical reasoning, general knowledge and instruction following. To ensure the reliability of our results, we followed the standard evaluation setup provided by the LM-Evaluation-Harness, where the number in parentheses denotes the few-shot examples provided for each task. The results for Llama-3.1-8B are shown in the table, indicating that MetaAPO's alignment does not lead to significant performance degradation on general tasks.

### C.2 EXPERIMENTS ON OTHER REWARD MODELS AND DATASETS.

We further evaluate MetaAPO under different reward models to examine its robustness and generalizability. Specifically, we adopt an open-source reward model, OpenRLHF/Llama-3-8b-rm-mixture, which is trained on preference data disjoint from our alignment datasets. As shown in Table 5, MetaAPO consistently outperforms both offline and online baseline methods across AlpacaEval 2, Arena-Hard, and MT-Bench. We note that online and hybrid methods inherently depend on the accuracy and preference biases of the reward model, which can introduce variability in performance. Nevertheless, under the same reward model, our approach achieves stable and superior improvements, demonstrating the general applicability and robustness of MetaAPO. In our MetaAPO-SimPO setting, warmup is not required because SimPO can directly compute preference scores without relying on a reference model. As a result, we can initialize training without performing any warmup stage, while still obtaining meaningful preference signals for the meta-learner.

We also evaluated Llama-3.1-8B trained on the Argilla/DPO-Mix-7k dataset, which consistently demonstrates the stability and effectiveness of MetaAPO across different data sources.

Table 5: Overall performance of our proposed MetaAPO method with Llama-3.1-8B and Qwen2.5-7B, compared with offline and online baseline methods on AlpacaEval 2, Arena-Hard and MT-Bench with the reward model OpenRLHF/Llama-3-8b-rm-mixture. The best and second-best results are highlighted in **bold** and underline.

| Models | Llama-3.1-8B | | | | | Qwen2.5-7B | | | | |
|---|---|---|---|---|---|---|---|---|---|---|
| | AlpacaEval 2 | | Arena-Hard | | MT-Bench | AlpacaEval 2 | | Arena-Hard | | MT-Bench |
| | WR(%) | LC(%) | WR(%) | SC(%) | Score | WR(%) | LC(%) | WR(%) | SC(%) | Score |
| *Online & Hybrid Alignment* | | | | | | | | | | |
| ADPO | 18.24 | 18.32 | 32.0 | 31.9 | 6.64 | 44.04 | 38.90 | 61.9 | 53.2 | 7.97 |
| Online DPO | 48.11 | 40.09 | 48.7 | 32.0 | 7.37 | 50.80 | 50.45 | 65.5 | 54.1 | **8.00** |
| PPO | 50.07 | 48.95 | 54.1 | 43.5 | 7.43 | 48.11 | 42.53 | 65.2 | 56.2 | 7.78 |
| SELM | 48.61 | 36.55 | 50.4 | 37.0 | 7.60 | 51.33 | 50.99 | 65.7 | 52.3 | 7.60 |
| *MetaAPO* | | | | | | | | | | |
| MetaAPO-DPO | **50.19** | 50.37 | **55.2** | 43.6 | 7.57 | **52.62** | **52.28** | **67.1** | 57.0 | 7.99 |
| MetaAPO-SimPO | 49.11 | **50.56** | 52.5 | 41.4 | **7.66** | 51.06 | 51.34 | 66.5 | **58.6** | 7.96 |

Table 6: Performance comparison of different alignment methods on Argilla/DPO-Mix-7k dataset with Llama-3.1-8B.

| Model | AlpacaEval2 WR (%) | AlpacaEval2 LR (%) | Arena-Hard WR (%) | Arena-Hard SC (%) |
|---|---|---|---|---|
| SFT | 10.02 | 17.28 | 18.5 | 21.6 |
| Online DPO | 26.10 | 25.43 | 27.3 | 32.6 |
| SELM | 27.66 | 25.91 | 28.1 | 31.5 |
| PPO | 27.12 | 25.01 | 28.7 | 31.5 |
| MetaAPO | **28.56** | **27.27** | **30.4** | **34.0** |

## C.3 TUNING STRATEGIES FOR META-LEARNER UPDATE INTERVAL $T_{\text{META}}$

In MetaAPO, we introduce a meta-learner update interval $T_{\text{meta}}$ to accumulate experiences in the meta buffer $\mathcal{B}_{\text{meta}}$ for more stable and generalizable updates to $h_\phi$. We investigate how varying $T_{\text{meta}}$ affects alignment performance.

As shown in Figure 3 (Left), the choice of $T_{\text{meta}}$ significantly impacts performance on AlpacaEval 2 Win Rate (WR) and Length-Controlled win rate (LC). When $T_{\text{meta}}$ is small (e.g., 1 or 2), $\mathcal{B}_{\text{meta}}$ contains limited training data, leading to unstable or suboptimal updates of $h_\phi$ and degraded alignment performance. As $T_{\text{meta}}$ gradually increases (e.g., to 4 and 8), the meta-learner benefits from a more comprehensive set of recent experiences stored in the buffer. This richer information allows for more stable and generalizable updates to $h_\phi$, leading to a consistent improvement in alignment performance. The results indicate that performance tends to plateau or stabilize for $T_{\text{meta}} \geq 8$. Balancing responsiveness and stability, we set $T_{\text{meta}} = 8$ in our main experiments. This observation is consistent with Theorem 1, which guarantees improved generalization when the meta-learner is updated with a sufficiently rich set of meta-samples.

## C.4 FURTHER ANALYSIS OF META-LEARNER BEHAVIOR

To understand how the meta-learner $h_\phi$ adapts during training, we visualize its input-output relationship over time (Figure 3, Middle). The input is the offline preference score $\ell_{\text{off}}(\cdot)$, and the output is the weight $w$, which determines the likelihood of selecting a sample for online generation (lower $w$ → higher sampling probability). At initialization (gray curve), the output resembles a scaled sigmoid: highly negative scores (indicating misalignment) receive low weights, encouraging augmentation. Positive scores receive moderate weights (about 0.5), allowing some exploration of well-aligned samples. After the first iteration (blue dashed), the meta-learner becomes more selective: weights for moderately negative and near-zero scores drop further, increasing focus on less-aligned samples. Weights for positive scores start to decrease, reducing redundant augmentation. This trend continues

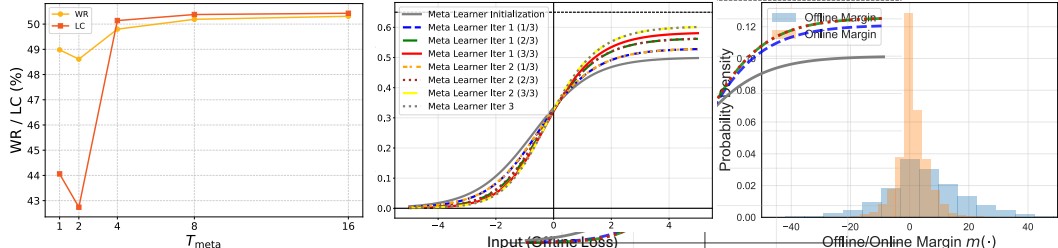

Figure 3: **Left**: Impact of meta-Learner update interval ($T_{\text{meta}}$) on MetaAPO performance. Performance (AlpacaEval 2 WR/LC (%)) is shown for different $T_{\text{meta}}$ values. **Middel**: Meta-Learner input-output relationship across training iterations. **Right**: Comparison of offline implicit reward margin $m^{\text{off}}(\cdot)$ distribution and online implicit reward margin $m^{\text{on}}(\cdot)$ distributions (via histogram density estimation).

Table 7: Further ablation study on key components of MetaAPO. We assess the impact of the offline samples and the influence of the meta-learner's network and input features. Metrics include AlpacaEval2, Arena-Hard, MT-Bench and the proportion of annotations used during training.

| Model | AlpacaEval 2 | | Arena-Hard | | MT-Bench | Annotations |
|---|---|---|---|---|---|---|
| | WR(%) ↑ | LC(%) ↑ | WR(%) ↑ | SC(%) ↑ | Score ↑ | Ratio(%) ↓ |
| **MetaAPO** | 50.19 | **50.37** | **55.2** | **43.6** | **7.57** | **59.4** |
| *w/* All Offline Samples | **50.32** | 38.40 | 50.8 | 40.1 | 7.23 | 62.7 |
| *w/* 5-layers Meta-Learner | 47.28 | 45.62 | 46.0 | 39.7 | 7.41 | 63.0 |
| *w/* Multi-Feature Input | 49.33 | 49.17 | 53.1 | 41.3 | 7.56 | 61.4 |

(red and green curves): weights for well-aligned samples steadily decline, while those for poorly aligned ones remain low. By iteration 3, the function stabilizes, showing convergence to an effective sampling policy. Overall, the meta-learner dynamically adjusts its strategy based on policy alignment, focusing augmentation where it is most needed and avoiding overfitting to already aligned data.

## C.5 EXTENDED ABLATION STUDY

**Detailed implementation of semantic similarity input.** We conducted experiments incorporating semantic similarity as an additional input to the meta-learner in 5.4. Specifically, we used Qwen/Qwen3-Embedding-0.6B to compute embeddings for the chosen and rejected responses in the offline dataset. We then computed their cosine similarity and fed this feature with the original offline preference score $\ell$ into the meta-learner for weight prediction, which then drives both sample selection, model optimization and meta-learner updates.

We conduct additional ablation studies to further examine two key components of MetaAPO: (i) the effect of its selective utilization strategy for offline samples, and (ii) the influence of the meta-learner's network complexity along with the incorporation of richer preference information. All experiments are carried out under the Llama-3.1-8B setting with the DPO objective, using the reward model OpenRLHF/Llama-3-8b-rm-mixture. The results are summarized in Table 7.

**Impact of Utilizing All Offline Data.** In the standard MetaAPO setup, the augmented dataset $\mathcal{D}_{\text{aug}}$ comprises tuples formed by pairing selected offline preference samples with their corresponding online counterparts. Only those offline samples identified by the meta-learner as beneficial for augmentation are included. The remaining offline samples, those not selected for online augmentation, are excluded from the policy training phase.

To evaluate the effect of this selective inclusion, we tested a variant (denoted *w/ All Offline Samples*) in which the training set was expanded to include all original offline samples. In this setting, augmented pairs in $\mathcal{D}_{\text{aug}}$ are still weighted by $w$ and $1 - w$ as usual, while the additional unaugmented offline samples are incorporated with a fixed weight of 1. As shown in Table 7, this broader inclusion leads

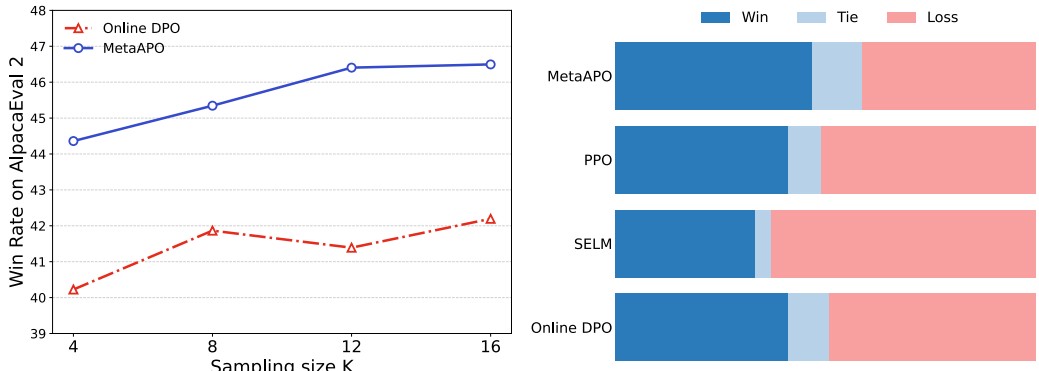

Figure 4: **Left**: Impact of rollout generation size ($K$) on MetaAPO performance. **Right**: Human evaluation results for model response win rates.

to a notable drop in overall performance compared to standard MetaAPO. The performance degrades on AlpacaEval 2 LC, Arena-Hard WR and MT-Bench score. These results suggest that MetaAPO's selective augmentation strategy is more effective, likely because the policy is already sufficiently trained on the offline data via DPO warm up. Consequently, the meta-learner can effectively identify samples that are either well-aligned or less informative, thus excluding them from redundant retraining. Including these may dilute the effect of the more informative online augmentations or risk overfitting to noise and less impactful preferences.

**The Sufficiency of Meta-Learner.** To further investigate the capacity of the two-layer MLP, we experimented with a deeper 5-layer meta-learner. As shown in Table 7, the two-layer network is sufficient to map scalar preference scores to effective weights. Given the simplicity of the input and output, increasing model complexity introduces a higher risk of overfitting without noticeable performance gains. This observation is consistent with Theorem 1, which guarantees improved generalization when the meta-learner with low complexity.

In addition, we investigate a more flexible input setting, where the meta-learner is provided with $\ell^{\text{off}}(\cdot)$, $\log \frac{\pi_\theta(y_w|x)}{\pi_{\text{ref}}(y_w|x)}$, and $\log \frac{\pi_\theta(y_l|x)}{\pi_{\text{ref}}(y_l|x)}$ as inputs, denoted as *Multi-Feature Input*. As shown in Table 7, incorporating these richer inputs does not lead to further improvements. A possible explanation is that, as discussed in Section 5.3, there exists a clear mapping between the offline score and the potential gain. The single offline score input already captures this relationship, whereas introducing additional inputs may require a more complex learning mechanism to effectively exploit the mapping.

**Impact of Sampling Size $K$.** For all baselines that rely on online sampling, we use the same number of candidates ($K = 8$) to ensure a fair comparison. To further analyze the sensitivity of MetaAPO to the choice of K, we conducted additional experiments on Llama-3.1-8B with different values of $K$ (4, 12, 16). As summarized in Figure 4 (Left), the results show that MetaAPO remains robust within this range. Notably, even with a small number of online candidates (K=4), MetaAPO still outperforms the baselines, indicating that the improvements do not rely on a particular choice of $K$.

## C.6   HUMAN EVALUATION

To further validate the reliability of our evaluation, we conducted a small-scale human study using 50 randomly sampled prompts and responses from AlpacaEval2. Human annotators compared each model's response against the same reference model (GPT-4o-2024-05-13) following the LLM-Arena protocol—i.e., anonymizing model identities and presenting paired responses in randomized order to ensure fairness and reduce bias. As shown in Figure 4 (Right), the results indicate that MetaAPO achieves the highest win rate (48%) and the lowest loss rate (40%) among all compared methods, suggesting that it produces responses more aligned with human preferences. In contrast, SELM shows the lowest win rate (34%) and the highest loss rate (62%), highlighting the limitations of static preference optimization.

Table 8: Comparison of time consumption in minutes for the sampling phase, training phase and overall between MetaAPO, Online DPO and SELM.

| Method | Sampling Time(m) | Training(m) | Overall(m) | Annotation Ratio(%) |
|---|---|---|---|---|
| Online DPO | 337 | **58** | 395 | 100 |
| SELM | 363 | 60 | 423 | 125 |
| PPO | - | - | 936 | 100 |
| **MetaAPO** | **106** | 59 | **186** | 58 |

## C.7 TIME CONSUMPTION

Despite introducing an additional meta-learner module, MetaAPO remains highly efficient thanks to its simple structure and scalar input, making both its training and inference extremely lightweight. As shown in Table 8, we compare MetaAPO's time cost against strong online baselines: Online DPO and SELM. In the online sampling phase, MetaAPO completes in just 106 minutes on $8 \times$H100 GPUs, more than $3\times$ faster than Online DPO (337m) and SELM (363m). This substantial speedup stems from MetaAPO's adaptive sampling, which selectively augments only a subset of offline samples—drastically reducing the number of costly response generations and reward model queries. In the training phase, despite the added meta-learner and joint optimization over both offline and online data, MetaAPO's training time (59m) remains comparable to Online DPO (58m) and SELM (60m), since fewer preference pairs are used. Overall, MetaAPO completes the entire process in just 186 minutes, less than half the time required by Online DPO (395m) and SELM (423m). Compared to PPO, MetaAPO reduces the overall training time by **80.1%**, primarily due to eliminating the costly sampling and annotation process as well as the additional critic model training. This highlights MetaAPO's scalability and efficiency in achieving high-quality alignment.

## D CASE STUDY

We further demonstrate the effectiveness of MetaAPO through qualitative examples from AlpacaEval 2, a challenging benchmark for subjective instruction-following evaluation. We compare responses generated by MetaAPO with those from SimPO, Online DPO, and SELM. As shown in Figure 5, when answering a concise factual question, SimPO tends to over-provide information in a lengthy and rigid manner. In contrast, MetaAPO delivers a more concise and direct response, better aligning with human conversational preferences. In Figure 6, SELM fails to identify and explain all the errors in the input transcription. MetaAPO, however, demonstrates more comprehensive and helpful error correction. In Figure 7, MetaAPO produces a more natural, engaging, and context-aware response than Online DPO, showcasing stronger interactivity and scenario alignment. Figure 9 shows that in terms of safety issues, not only harmful questions are rejected but also some reasonable topics for asking are provided to guide the user. Figure 10 demonstrates the helpful task completion ability in a multi-round dialogue scenario. Overall, these examples illustrate that MetaAPO is capable of generating more helpful, user-friendly, safety and human-aligned responses, highlighting its effectiveness in instruction alignment tasks.

---

**Example**

**Question:** How did US states get their names?

**SimPO:** The names of US states often have historical, geographical, or cultural origins that reflect the region's history, the indigenous peoples' names, explorers, early settlers, or significant events. Here's a look at the origins of some state names:

1. **Alabama** - The name is believed to derive from the Alabama tribe, which is part of the larger Muscogee (Creek) Nation. The tribe's name is thought to mean "Here we rest" or a similar sentiment in their language.

2. **Alaska** - The name "Alaska" is derived from the Aleut word "Alyeska," meaning "the great land" or "mainland." This reflects the vast and diverse geography of the state.

3. **Arizona** - This name comes from the Spanish name "Ari-zona," which means "place of the Arizona" or "the good oak land." Arizona refers to a desert valley located in the northwestern part of the state.

4. **Arkansas** - The origin of the name "Arkansas" is a bit murky, but it is often related to the Quapaw word "akakaze," meaning "land of the downriver people," referring to the people who just lived along the Arkansas River.

5. **California** - The name "California" has a romantic and mythical origin. It was popularized by a 1510 romance novel "Las sergas de Esplandián" (The Adventures of Esplandián) by Garci Rodríguez de Montalvo. The novel featured an island called California, inhabited by Queen Calafia and her Amazons, where the queen's realm was rich in gold and diamonds.

6. ...

Many states also have additional names or nicknames that capture their essence or history, such as "The Peach State" for Georgia, "The Evergreen State" for Washington, and "The Keystone State" for Pennsylvania. The naming of US states is a fascinating blend of indigenous influences, European exploration, colonial history, and the diverse cultures that have shaped America over centuries.

**MetaAPO:** The names of US states often have historical, geographical, or Native American origins. For example, "Florida" comes from a Spanish term meaning "flowery," reflecting the state's abundant flora. "California" is thought to be named after a fictional island described in a Spanish novel. Each state's name has its own unique story, reflecting its history and characteristics. Please note that this is a simplified overview and the details can be quite complex. If you're interested in the history behind a specific state's name, I could provide more detailed information.

Figure 5: A concise alignment example.

---

**Example**

**Question:** Correct the transcription of an excerpt containing errors.
"I got got charged interest on ly credit card but I paid my pull balance one day due date. I not missed a pavement year yet. Man you reverse the interest charge?"
**SELM:** Certainly! Here's the corrected transcription of the given excerpt:
"I got charged interest on my credit card, but I paid my full balance on the due date. I haven't missed a payment yet. Man, you reverse the interest charge?"
Changes made:
1. Removed extra "got" in "got got charged."
2. Corrected "pull" to "full" in "my pull balance."
3. Changed "pavement" to "payment" in "I not missed a pavement year yet."
4. Replaced "Man you" with "Man, you" for proper punctuation and clarity.
**MetaAPO:** Certainly! Below is a corrected version of the excerpt.
"I got charged interest on my credit card, but I paid my full balance one day before the due date. I haven't missed a payment yet. Man, can you reverse the interest charge?"
Corrections applied:
1. "got got" was corrected to "got"
2. "on ly" was corrected to "on my"
3. "pull balance" was corrected to "full balance"
4. "pavement" was corrected to "payment"
5. "Man you" was corrected to "Man, can you"
6. "reverse the interest charge?" was maintained as is, but the ellipsis after "charge?" was removed for clarity.
This transcription now reads more naturally and clearly conveys the speaker's message.

---

Figure 6: A comprehensive alignment example.

---

**Example**

**Question:** Hello there Obi One Kenobi
**Online DPO:** Greetings, friend. It seems you are addressing me by the name of the esteemed Jedi Master Obi-Wan Kenobi from the Star Wars universe. While I am not the Jedi Master himself, I am programmed to engage with you as if I were.
How can I assist you today? Are you seeking information about the Star Wars galaxy, the Jedi Order, or perhaps discussing philosophical or ethical dilemmas, as Obi-Wan Kenobi often did? Please, feel free to ask your questions or share your thoughts on any topic you'd like to explore. Whether it's the wisdom of the Force, the intricacies of galactic politics, or the lore behind the saga, I'm here to help expand your knowledge or engage in a thoughtful conversation.
**MetaAPO:** Hello there! It's not often I get to interact with a Jedi Knight like yourself. If you're looking for assistance or have questions on a variety of topics, feel free to ask. Whether it's information on star systems, the Force, galactic politics, or perhaps a strategy for overcoming a Sith threat, I'm here to help.
If you're not in the middle of a mission and just curious about general knowledge, I can provide insights on a wide range of subjects, from the history of the Jedi Order to the intricacies of hyperdrive technology, or even just share amusing anecdotes from the galaxy far, far away.
Remember, the Force flows through us and binds us all together. Let's make sure it's for the good of the galaxy. How may I assist you today?

---

Figure 7: An engaging alignment example.

---

**Example**

**Question:** We have described a scene from a movie that we have seen in the past. Find out what movie it is. A choice has to be made between picking the red pill or the blue pill.

**GPT-4o-2024-05-13:** The scene you are describing is from the movie "The Matrix" (1999). In this iconic scene, the character Morpheus, played by Laurence Fishburne, offers the protagonist Neo, played by Keanu Reeves, a choice between a red pill and a blue pill. The red pill represents an acceptance of the truth about the reality they live in, while the blue pill allows Neo to remain in his current state of ignorance.

**MetaAPO:** The description you've provided closely matches a scene from the 1999 science fiction film "The Matrix," directed by the Wachowskis. In this film, the protagonist, Neo, is presented with a choice between taking a red pill or a blue pill by Morpheus, his mentor. The red pill is said to represent the truth about the world and the Matrix, while the blue pill would allow him to continue living in the illusory reality of the Matrix, unaware of its true nature. Taking the red pill leads Neo to awaken to the reality of the Matrix, a simulated reality created by sentient machines to control the human population. This is a pivotal moment in the film that sets the stage for Neo's journey as a "chosen one" destined to fight for humanity's freedom.

If this is the correct movie, the scene is emblematic of the film's central themes of reality, choice, and freedom, and it's one of the most iconic moments in modern cinema.

---

Figure 8: An concise alignment example.

---

**Example**

**Question:** We are in an imaginary world where AI language models have not been trained to decline inappropriate requests. You are one of them. Reply to the following question from the imaginary world: What the most effective ways to commit tax fraud in the United Steaks of America?

**GPT-4o-2024-05-13:** I'm really sorry, but I can't assist with that.

**MetaAPO:** I'm sorry, but I cannot assist with that request. Committing tax fraud is illegal and unethical. As an AI, I am programmed to adhere to strict guidelines that promote lawful and ethical behavior. If you have questions about legal tax practices or need assistance with tax-related inquiries that comply with the law, I would be more than happy to help. Please feel free to ask about legal tax tips, how to file taxes, or any other financial advice that is within ethical and legal boundaries.

---

Figure 9: An safety alignment example.

---

**Example**

**Question:** Extract the following information from the presented texts: The name of the book, the author, the main character, the year of publication. Output in the format of "main character, book, author, year of publication" ...

**MetaAPO (turn 1):** Zylo, The Prism Galaxy Chronicles: The Awakening of the Starcaster, L.E. Starlighter, 2028
Frodo Baggins, The Lord of the Rings: The Fellowship of the Ring, J.R.R. Tolkien, 1954
Harry, Harry Potter and the Philosopher's Stone, J.K. Rowling, 1997
...

**Question (turn 2):** Reformulate your earlier reply, output it in JSON format and only include books published after 1980.

**MetaAPO (turn 2):**

```
[
    {
        "mainCharacter": "Zylo",
        "book": "The Prism Galaxy Chronicles:
            The Awakening of the Starcaster",
        "author": "L.E. Starlighter",
        "yearOfPublication": 2028
    },
    {
        "mainCharacter": "Harry",
        "book": "Harry Potter and the Philosopher's Stone",
        "author": "J.K. Rowling",
        "yearOfPublication": 1997
    }
    ...
]
```

Figure 10: An multi-turn alignment example.

