# OpenReview forum: "Alignment through Meta-Weighted Online Sampling: Bridging the Gap between Data Generation and Preference Optimization"
_ICLR.cc/2026/Conference — ICLR 2026 Poster_

### Official Review · Reviewer_Xifs · 2025-10-24

**Soundness:** 2
**Presentation:** 3
**Contribution:** 2
**Rating:** 4
**Confidence:** 4

**Summary:**

This paper addresses the distribution mismatch of the offline alignment dataset and the evolving model state in preference optimization.
The authors propose Meta-Weighted Adaptive Preference Optimization (MetaAPO), a framework that adaptively adjusts the online sampling and training sampling weights based on the model's current state.
To achieve this, they utilize a meta-learner (two-layer MLP) that takes the model's preference score on each sample as input to determine whether to augment the sample with online sampling and to adjust the training sampling weight accordingly.
The training of this meta-learner encourages it to improve the overall preference score of the current model on the training samples (including both online and offline samples).
Experiments on different models and benchmarks demonstrate the effectiveness of MetaAPO in improving preference optimization performance over existing static and online methods.

**Strengths:**

1. **Clear Motivation and Simple Solution**: Solving the distribution mismatch between offline alignment data and the evolving model state is a well-motivated problem in direct preference optimization scenarios, and MetaAPO provides a straightforward adpative sampling and weighting solution to address this issue.
2. **Writing Quality**: The paper is well-written and easy to follow, with clear explanations of the proposed method and experimental results.

**Weaknesses:**

1. **Concerns on Reasonability**: The preference score $\ell$ input to the meta-learner measures the agreement between the model and the human preference data, with a higher score indicating a lower DPO loss and alignment with human preferences.
The objective of the meta-learner is to improve the overall preference score of the model on training samples, so data samples that the model already agrees with (high $\ell$) are less likely to be augmented with online sampling and are assigned higher training weights. This seems counterintuitive, as one would expect the model to focus more on samples it currently fails to align with (low $\ell$) to improve alignment [1,2]. The authors should provide more justification for this design choice.
2. **Limited Ablation Studies**: As shown in Figure 3, the learned meta-learner's mapping from preference score $\ell$ to weight $w$ appears to be similar across different iterations. This raises the question of whether a fixed sampling/loss weighting scheme could achieve similar performance. Although the ablation study "w/ uniform loss weighting" and "w/o meta-learner" are provided, a loss weighting/ sampling scheme based on the final learned meta-learner (fixed after training) should be included for a more direct comparison.
3. **Baseline Comparisons**: It seems like the ablation "w/ all sampling" is equivalent to the "online DPO" baseline? The authors should clarify the difference between these two settings to avoid confusion. Moreover, the authors should consider including more data filtering baselines in the online sampling setting, such as [2,3,4] (which are already cited in the related work section), to provide a more comprehensive comparison since they also take the model state into account when selecting samples for preference optimization.

**References**\
[1] Yang, Sen, et al. "Not All Preference Pairs Are Created Equal: A Recipe for Annotation-Efficient Iterative Preference Learning." Findings of the Association for Computational Linguistics: EMNLP 2024. 2024.\
[2] Huang, Kexin, et al. "Larger or Smaller Reward Margins to Select Preferences for LLM Alignment?." Forty-second International Conference on Machine Learning.\
[3] Deng, Xun, et al. "Less is more: Improving llm alignment via preference data selection." arXiv preprint arXiv:2502.14560 (2025).\
[4] Gao, Chengqian, et al. "Principled Data Selection for Alignment: The Hidden Risks of Difficult Examples." Forty-second International Conference on Machine Learning.

**Questions:**

Can you provide more ablation studies comparing to fixed sampling/loss weighting schemes based on the learned meta-learner after training, as mentioned in Weakness 2?

---

> ### Author Response · Authors · 2025-11-17
> **Responses to Reviewer Xifs (Part 1)**
>
> Thank you for your valuable comments. We hope our responses below can help address your concerns on this paper.
>
> > Q1: Explanation of meta-learner score assignment
> >
>
> Thank you for the question. We would like to clarify that **MetaAPO is designed to address the currently misaligned samples**. As noted by the reviewer, the input score $\ell$ reflects the agreement between the current model and human preference data.
>
> Samples with **lower scores** correspond to harder or potentially OOD cases. Recent data-selection works show that such low-score samples can destabilize training and are often **discarded entirely** [1,2,3], leaving only the high-score subset for optimization.
>
> MetaAPO takes a different view: these low-score samples are **exactly where alignment gaps occur, and can be effectively corrected through dynamic online augmentation** rather than being removed.
>
> Concretely (as described in Section 4.1):
>
> - **For low-score samples**: The meta-learner assigns a small weight $\omega$, which (i) **increases the probability of being sampled** for online augmentation, and (ii) shifts optimizing away from offline samples ($\omega$) towards online feedback ($1 - \omega$).
> - **For high-score samples**: These receive larger weights and thus have a **low probability of being sampled** again, which allows the model to reduce the cost of online sampling and maintain stability on already aligned offline data.
>
> This behavior is also **empirically reflected in Figure 2 (middle)**, where sampled points concentrate in the low-score region (orange points toward the left on the x-axis), while high-score samples are rarely selected.
>
> In summary, MetaAPO **targets the samples that are currently misaligned**, enabling efficient correction of OOD cases while avoiding unnecessary updates on already aligned examples.
>
> References:
>
> [1] Gao et al. Principled Data Selection for Alignment: The Hidden Risks of Difficult Examples, ICML, 2025.
>
> [2] Deng et al. Less is more: Improving llm alignment via preference data selection, NeurIPS, 2025.
>
> [3] Ko et al. SERA: Self-rewarding and Alignment of LLMs Using Implicit Reward Margins, ICLR, 2025.
>
> > Q2: More ablation studies about fixed sampling/loss weighting
> >
>
> We have added **three new ablation studies** to further verify the necessity of *dynamic* learning in MetaAPO:
>
> - **Fixed Sampling and Loss Weighting**: We initialize the meta-learner using the weights learned by MetaAPO and **freeze it entirely**, keeping both sampling weights and loss weights fixed throughout training.
> - **Fixed Sampling**: We initialize two meta-learners with the learned weights, **freeze the sampling meta-learner**, and allow the loss-weighting meta-learner to continue updating dynamically.
> - **Fixed Loss Weighting**: Similarly, we freeze the loss-weighting meta-learner while allowing the sampling meta-learner to update dynamically.
>
> |  | **AlpacaEval2 WR** | **AlpacaEval2 LR** | **Arena-Hard WR** | **Arena-Hard SC** | **MT-Bench** |
> | --- | --- | --- | --- | --- | --- |
> | MetaAPO | **47.48** | **43.21** | **43.9** | **40.8** | **7.56** |
> | w/ Fixed Sampling and Loss Weighting | 45.01 | 40.03 | 39.9 | 36.4 | 7.22 |
> | w/ Fixed Sampling | 45.90 | 42.04 | 40.2 | 37.1 | 7.34 |
> | w/ Fixed Loss Weighting | 46.69 | 41.19 | 40.4 | 37.6 | 7.38 |
>
> We have included the above results in **Table 2** in the revised paper.
>
> > Q3: Explanation of the ablation "w/ all sampling" setting
> >
>
> Thank you for your comment. We would like to clarify that “*w/ all sampling”* in our ablation study refers to **removing** the meta-learner’s adaptive online sampling mechanism (introduced in Section 4.1) and directly applying online sampling to all samples. However, the meta loss weighting and the update of the meta-learner **still follow** the MetaAPO design, which differs from the policy optimization procedure used in Online DPO.
>
> We have revised **Section 5.4** and added further explanation accordingly.

---

> > ### Author Response · Authors · 2025-11-17
> > **Responses to Reviewer Xifs (Part 2)**
> >
> > > Q4: Comparison with more recent baselines
> > >
> >
> > We further incorporated several recent baseline methods into our comparison to provide a more comprehensive comparison under the online sampling setting. The results are shown in the table below.
> >
> > Specifically, **DPO-Shift** [1] introduces a re-weighting strategy for negative samples but shows only marginal gains over standard DPO. We also included **BeeS** [2] and **MAP** [3], both of which enhance preference optimization by selecting or filtering higher-quality data using their respective sampling heuristics. (**Selective DPO** [4] was already included in our original submission.)
> >
> > For all methods, we used the **default thresholds** and **hyperparameters** reported in their original papers to ensure faithful reproduction. A detailed description of these baselines and our implementation has been added to **Appendix B.3**.
> >
> > Across all metrics, MetaAPO consistently outperforms these baselines. Please refer to **Table 1** for complete results.
> >
> > |  | **AlpacaEval2 WR** | **AlpacaEval2 LR** | **Arena-Hard WR** | **Arena-Hard SC** | **MT-Bench** |
> > | --- | --- | --- | --- | --- | --- |
> > | MetaAPO | 47.48 | 43.21 | 43.9 | 40.8 | 7.56 |
> > | DPO-Shift | 19.33 | 19.39 | 28.7 | 27.6 | 7.03 |
> > | Selective DPO | 21.35 | 20.67 | 28.5 | 31.9 | 7.11 |
> > | BeeS | 41.42 | 37.45 | 39.5 | 36.4 | 7.13 |
> > | MAP | 43.60 | 38.97 | 40.4 | 37.2 | 7.17 |
> >
> > References:
> >
> > [1] Yang et al. DPO-Shift: Shifting the Distribution of Direct Preference Optimization, Arxiv, 2025.
> >
> > [2] Deng et al. Less is more: Improving llm alignment via preference data selection, NeurIPS, 2025.
> >
> > [3] Huang et al. Larger or Smaller Reward Margins to Select Preferences for LLM Alignment?, ICML, 2025.
> >
> > [4] Gao te al. Principled Data Selection for Alignment: The Hidden Risks of Difficult Examples, ICML, 2025.

---

> ### Author Response · Authors · 2025-11-24
>
> Dear Reviewer Xifs,
>
> We greatly appreciate the time and effort you've taken to review our submission. We hope that our response has addressed the concerns raised in your initial reviews, and we look forward to your feedback.
>
> Please let us know if you need any further information or clarification.
>
> Best regards and thanks,
>
> Paper 12533 Authors

---

> > ### Comment · Reviewer_Xifs · 2025-11-24
> >
> > I appreciate the authors' detailed responses to my previous questions and their efforts to clarify the design choices and experimental results. However, I still have some concerns about the experiment design and the learned meta-learner functions:
> >
> > 1. **On the design choice of the meta-learner’s objective (related to Weakness 1)**:\
> > I agree that for high-score samples, it is reasonable to avoid redundant sampling and unnecessary training.
> > However, for low-score samples, you mentioned that one benefit of promoting online sampling is to reduce OOD issues.
> > In settings such as SimPO (and its follow-up work), where the responses in $D_{offline}$ are collected using the same model that is later optimized, does the OOD problem for low-score samples still persist?
> > 2. **On the learned mapping functions of the meta-learner**:
> >     - I am still unsure why fixed weighting/sampling schemes perform clearly worse than the adaptive meta-learner, given that the learned mappings in Figure 3 appear quite stable across iterations. Does the mapping actually change in a more fine-grained manner during training?
> >     If possible, could you provide the trajectory of the learned function at a finer temporal resolution (e.g., per optimization step rather than once per iteration as in Figure 3)?
> >     - When optimizing the meta-learner (Eq.6), are the preference scores $\ell$ for online and offline samples on comparable scales?
> >     The preference scores on online samples might be significantly smaller than that on offline samples.
> >     If not, did you apply any normalization to them before calculating the loss?
> >     - Finally, in Figure 3, all learned weights lie below 0.6 rather than spanning the full [0,1] range. Is this due to an intentional range restriction, or is it simply what the model converged to during training?

---

> ### Author Response · Authors · 2025-11-25
> **Responses to Reviewer Xifs (Part 1)**
>
> We sincerely thank the reviewer for the additional detailed feedback. Below, we address your questions regarding the OOD of offline data, the fine-grained dynamics of the meta-learner, and the technical details of score scaling and weight ranges.
>
> > Q1: On the OOD problem and low-score samples
>
> 1. **The Persistence of OOD in SimPO Settings.** The OOD problem persists in SimPO, primarily due to the temporal mismatch between the **static** offline dataset and the **evolving** policy during training.
>    - **Static vs. Evolving:** Even if the offline data was generated by the base model initially, the dataset remains pre-collected and **static** while the policy $\pi_\theta$ is continuously updated during training. As the policy evolves, its distribution naturally diverges from the fixed offline samples.
>    - **Mismatch of Low-Score Samples:** The existence of distribution mismatches and noise in self-generated data is further evidenced by recent works such as BeeS [1] and MAP [2]. They also employ heuristic metrics and fixed thresholds to filter out low-score/noisy subsets to enhance training. We have added these methods as baselines in our revised **Table 1**  to demonstrate that MetaAPO outperforms such heuristic filtering strategies.
> 2. **Practical Inconvenience of Pure Self-Generation.** Beyond the OOD issue, relying entirely on self-generated data for every specific model version is **practically burdensome and inefficient**. Unlike offline datasets, which are reusable and transferable across different experiments, a pure self-generation approach requires computationally expensive sampling and annotation for each new model architecture or checkpoint. Instead of generating the entire dataset, MetaAPO employs a meta-learner to identify specific samples within the offline dataset that require further online augmentation.
> 3. **Motivation for MetaAPO's Hybrid Approach.** To effectively address practical challenges and the distribution shift in preference alignment, we adopt a hybrid approach:
>    - **Value of Offline Data:** We leverage high-quality open-sourced offline preference datasets because they are readily accessible, diverse and cost-efficient.
>    - **Goal:** Our primary goal is to determine how to best leverage large, pre-existing offline preference datasets while addressing the significant issue of distribution shift during alignment. We aim to achieve this with minimal reliance on costly online preference augmentation.
>
> **References:**
>
> [1] Deng et al. Less is more: Improving llm alignment via preference data selection, NeurIPS, 2025.
>
> [2] Huang et al. Larger or Smaller Reward Margins to Select Preferences for LLM Alignment?, ICML, 2025.

---

> > ### Author Response · Authors · 2025-11-25
> > **Responses to Reviewer Xifs (Part 2)**
> >
> > > Q2: On the learned mapping functions of the meta-learner
> > >
> > 1. **Fine-grained Trajectory Visualization.** Thank you for your suggestion. We visualized the trend of the meta-learner in a more fine-grained manner (every 1/3 of an iteration) in **revised Figure 3 (Right)**.
> >     - **Observation:** As shown in the figure, we observe a **"Reset-and-Climb"** cyclic pattern. Specifically, at the transition between iterations (e.g., Iter 1 End $\to$ Iter 2 Start), the mapping curve shifts **downward** (from Red to Orange dashed). Subsequently, as training progresses within Iteration 2, the curve progressively shifts **upward** (Orange $\to$ Brown $\to$ Yellow).
> >     - **Interpretation of Dynamic Optimization.** This trajectory reflects a fine-grained dynamic optimization schedule assigned by the meta-learner within a iteration:
> >         - **Phase 1: Adaptation (Curve Down-shift):** At the beginning of a new training iteration (e.g., Iter 2, 1/3), the meta-learner lowers the weights assigned to offline samples (for $\ell^\text{off} > 0$). This effectively increases the relative weight of the online component $(1-w)$ in the loss function. This indicates that immediately after generating fresh, on-policy data, the model prioritizes **"digesting"** this new feedback to correct immediate alignment errors.
> >         - **Phase 2: Anchoring (Curve Up-shift):** As training within the iteration nears completion (e.g., Iter 2, 3/3), the meta-learner increases the offline weights $w$. This shifts the focus back to the offline component, acting as a **regularization mechanism**. It "anchors" the policy back to the high-quality offline annotations to prevent catastrophic forgetting or overfitting to the model's own generations.
> >     - **Superiority over Fixed Schemes.** This visualization explains why fixed weighting schemes perform worse. Static schemes apply constant pressure, lacking the capability to dynamically toggle between plasticity (learning from new feedback via low $w$) and stability (consolidating prior knowledge via high $w$). MetaAPO automatically learns this pattern, which static heuristics cannot achieve.
> > 2. **The offline and online preference score scaling.** The preference scores are on the same scale and no external normalization is required.
> >     - **Basis:** As defined in Eq. 4, both $\ell^\text{off}$ and $\ell^\text{on}$ are calculated using the **same current policy** $\pi_{\theta}$ and the **same reference model** $\pi_{ref}$. They represent log-likelihood ratios within the same mathematical framework.
> >     - **Empirical Evidence:** We quantitatively analyzed the distribution of the score gap $(\ell^\text{on} - \ell^\text{off})$ shown in **Figure 2 (Middle)**. The gap follows a near-symmetric distribution centered close to 0 (Mean $\approx$ -1.56), and the range of $\ell^\text{on}$ effectively overlaps with $\ell^\text{off}$. This confirms that both scores operate on the same magnitude, validating the direct subtraction in the gradient estimation without normalization.
> > 3. **The convergence of meta-learner.** The weights of the meta-learner are the result of natural convergence during training. We did not apply any clipping or artificial restrictions. The model learns a "soft balance", assigning significant weight to offline data for stability (about 0.6) while reserving probability mass for online exploration (the remaining 0.4). This aligns with the "explore-integrate" behavior described in Section 5.3.
> >
> > Thank you again for taking the time to review our paper. Your comments have helped us improve the quality of the paper and we truly appreciate your feedback. If you have further questions, please feel free to let us know.

---

> > > ### Comment · Reviewer_Xifs · 2025-11-25
> > >
> > > Thanks for your detailed and helpful answers.
> > > In Figure 3 (the updated version), the changing mappings address my previous concern about fixed weighting/sampling schemes.
> > >
> > > However, I still require clarification concerning the interpretation of Figure 2 (middle).
> > >
> > > You state that the figure demonstrates *"the range of $\ell^\text{on}$ effectively overlaps with $\ell^\text{off}$"*.
> > > But based on the current visualization, where the axes of the figure are $\ell^\text{on} - \ell^\text{off}$ (not $\ell^\text{on}$), and $\ell^\text{off}$, I am unable to observe this overlap as you described.
> > >
> > > In fact, the scattered points appear to be strongly correlated and can be approximately fitted by a line like $y = -x$.
> > > If $\ell^\text{off}$ has a significantly larger scale than $\ell^\text{on}$, such that $\ell^\text{on} - \ell^\text{off} \approx -\ell^\text{off}$, then such a strong negative correlation is expected.
> > > Could you please clarify this point further?
> > > For example, providing the mean and variance of $\ell^\text{on}$ and $\ell^\text{off}$ separately would help.

---

> ### Author Response · Authors · 2025-11-26
>
> We sincerely thank the reviewer for these insightful observations. Below we provide further clarification regarding the scale of the preference scores, which we hope will resolve your concerns.
>
> 1. **Empirical Results:** We computed the mean and standard deviation of both the offline and online scores across **all samples** to ensure stable and accurate statistics. The results are:
>
>     **Offline Score** (Mean: –0.6765, Std: 0.6497) and **Online Score** (Mean: –0.6762, Std: 0.2476).
>
>     We also visualized their distributions using KDE estimation in **Figure 3 (Right)**. The results show that the means of the two scores are nearly identical. The offline score exhibits a more long-tailed distribution, but the two largely overlap in their main support regions. There is no notable difference in their numerical scale.
>
> 2. **Further Analysis:**
>     1. The two scores share the **same physical meaning and scale**: both are computed using the same model and same prompts, **ensuring consistent evaluation criteria**. Since both the mathematical definition and the model parameters are identical for offline and online data, the resulting scores lie in the same metric space. **Their values directly reflect the absolute implicit reward relationship.**
>     2. Since the offline dataset consists of diverse sources and heterogeneous answer qualities, its score distribution is broader (as shown in Figure 3 (Right)). In contrast, online data are generated by the model itself, leading to more stable scores. The difference in standard deviation induces the negative correlation observed in **Figure 2 (Middle)**. For instance, low-quality offline samples yield significantly worse scores compared to the stable online model outputs, prompting the meta-learner to assign more weight to online data. Conversely, for high-quality offline samples, the online model may not yet match their quality, leading the meta-learner to rely more on offline data. We leverage this relative relationship to assess the condition of the model with respect to offline samples, whereas prior works relied on manually specified thresholds for such decisions.
> 3. **Example:** Suppose an offline sample has poor quality with a score of –2.5, while an online sample achieves –1.5, indicating that online generation could provide useful improvement. However, if we separately normalize each distribution, the offline score may become 0.2 while the online score becomes 0.1. In that case, the meta-learner would fail to recognize that the online data is actually much better than the offline sample, leading to incorrect weight assignment.
>
> Thank you again for your effort and valuable feedback in helping us improve the quality of our paper.

---

> ### Comment · Reviewer_Xifs · 2025-11-26
>
> While I appreciate the authors' efforts to clarify my previous questions, I find there are several unclear points regarding either the analysis or the experimental design.
> I strongly suggest that the authors should keep a consistent notation and comparable analysis to avoid confusion:
>
> 1. When you refer to *"all samples,"* for Figure 3 (right, updated version), does this imply a different subset than the one used in Figure 2 (middle)?
> 2. In Figure 3 (right, updated version), why does the preference score contain positive values? $\log \sigma(\cdot)$ should always be negative.
> If this problem is due to the KDE estimation, please use a more appropriate visualization method to avoid confusion.
> 3. Why do you use $\log r_w - \log r_l$ but still calling it preference score $\ell = \log \sigma(r_w - r_l)$ in Figure 2 ($r$ denotes the implicit reward)?
> Inconsistent notation undermines the rigor of the analysis. \
> Moreover, can you provide the mean and variance of this $\log r_w - \log r_l$ for online and offline samples separately, to directly verify whether their ranges are different, as I suspected before?
> 4. As you've mentioned: *"offline dataset consists of diverse sources and heterogeneous answer qualities ... online data are generated by the model itself, leading to more stable scores"*.
> This is exactly why I asked whether the OOD issue still exists in SimPO-like settings in my first follow-up questions.
>
> My major concern with this work is whether the experiment design is reasonable and general enough:
> - The current pipeline requires a warm-up training on the offline dataset. This process will clearly make the model's preference scores on the offline dataset contain some additional information. It is unclear if MetaAPO remains effective without this specific preconditioning.
> - In SimPO-like settings, where the offline data is collected using the same model that is later optimized, the distribution of offline and online data will be much closer compared to the current experimental settings.
> Therefore, the performance of MetaAPO in such scenarios is still unclear to me.
>
> I think the authors should discuss these concerns and provide empirical evidence to support the generality of MetaAPO in these common settings.

---

> ### Author Response · Authors · 2025-11-28
> **Responses to Reviewer Xifs (Part 1)**
>
> We sincerely appreciate the reviewer’s patient and detailed feedback. We provide additional empirical evidence and explanations below to address your concerns.
>
> **The Rationality and Universality of Experimental Design**
>
> 1. Discussion on initial warmup
> - **Setting Rationality:** Initial warmup alignment is a commonly adopted strategy to bootstrap preference models with offline data and provides a preference score that reflects how well the current model aligns with offline preferences. This warmup setting has been also used in recent alignment works [1–4]. Our method builds upon this initial alignment by introducing a meta-learner that adaptively learns a mapping between the offline alignment signal and the potential gain from online exploration, instead of relying on a fixed rule or threshold that places reliance on absolute scores.
> - **Comparison Fairness:**
>     - In Table 1, for all baselines that depend on the preference scores, such as Selective DPO [1] and BeeS [2], **we apply the same warmup procedure to ensure a fair comparison**.
>     - Our paper also provides comparisons with baseline methods trained on hybrid data in Table 1, further confirming that MetaAPO’s effectiveness.
> - **Robustness:**
>     - **Less warmup data**: We also investigate reducing the amount of data used for DPO score warmup. As shown in Table below, MetaAPO uses only 50% of the offline data for initial alignment still yields effective preference signals. This demonstrates that the meta-learner can maintain stable performance and continually refine the mapping based on DPO scores as training progresses, reducing reliance on extensive initial alignment while retaining adaptability.
>     - **Without warmup**: We can directly compute the SimPO score **without warmup**, and as shown in the table below, it still achieves comparable results. **In the original paper's Appendix C.2 and Table 5**, we presented the results of MetaAPO using SimPO (denoted as MetaAPO-SimPO) to calculate preference scores under the supervision of additional reward model, which did not involve warmup. We have revised the paper to more clearly describe this experimental setup (Appendix C.2).
>
> |  | **AlpacaEval2 WR**  | **AlpacaEval2 LR** | **Arena-Hard WR** | **Arena-Hard SC** |
> | --- | --- | --- | --- | --- |
> | MetaAPO (w/ 50% warmup data) | 46.13 | 42.10 | 42.5 | 39.5 |
> | MetaAPO (w/ SimPO score) | 46.46 | 43.07 | 43.6 | 41.0 |
> | MetaAPO | 47.48 | 43.21 | 43.9 | 40.8 |
>
> **References:**
>
> [1]  Gao et al. Principled Data Selection for Alignment: The Hidden Risks of Difficult Examples, ICML, 2025.
>
> [2] Deng et al. Less is more: Improving llm alignment via preference data selection, NeurIPS, 2025.
>
> [3] Ko et al. SERA: Self-rewarding and Alignment of LLMs Using Implicit Reward Margins, ICLR, 2025.
>
> [4] Chen et al. Bootstrapping Language Models with DPO Implicit Rewards, ICLR, 2025.

---

> ### Author Response · Authors · 2025-11-28
> **Responses to Reviewer Xifs (Part 2)**
>
> 2. MetaAPO under SimPO-like settings
>
> We further conducted experiments using self-generated data as the offline dataset.
>
> - **Experimental setup and results:** Specifically, we employed the Llama-3.1-8B SFT model to generate samples and form a preference dataset. During the training phase, this dataset was used to replace the offline Ultrafeedback dataset originally utilized in the paper. Additionally, to eliminate the impact of warmup, we continued to use SimPO for calculating preference scores. The experimental results are shown in the table below. MetaAPO (SimPO-like setting) achieves the comparable results and outperforms other baselines.
> - **Discussion on offline samples source:** Our goal is to leverage the meta-learner to identify potential OOD samples within the offline dataset and improve them through online enhancement. **Our method does not impose any strict constraints on the source of offline data**. In our paper, for implementation simplicity, we used pre-collected open-source preference datasets as the offline data. This setup is common in preference optimization. Compared with SimPO-like approaches, open-source offline data avoids the substantial cost of generation and annotation.
> - **Further discussion:**
>     - In Simpo-like settings, using data generated by the model itself is closer to on-policy data compared to open-source offline data. However, due to the static nature of the data and the continuous training of the model, the issue of OOD still persists (where the model's preference score does not align with the dataset's preferences).
>     - For DPO scores or SimPO scores, the same set of criteria is used to calculate $\ell_\text{off}(\cdot)$ and $\ell_\text{on}(\cdot)$ when the model scores offline and online data, so the absolute values can be directly compared.
>
> |  | **AlpacaEval2 WR**  | **AlpacaEval2 LR** | **Arena-Hard WR** | **Arena-Hard SC** |
> | --- | --- | --- | --- | --- |
> | MetaAPO (SimPO-like setting) | 46.04 | 42.37 | 42.6 | 40.0 |
> | MetaAPO | **47.48** | **43.21** | **43.9** | **40.8** |
> 3. Summary of additional experimental settings
> - **Different models and baselines.** In Table 1, we evaluate MetaAPO on two mainstream model families (Llama and Qwen) and compare it against offline, online and hybrid preference-optimization methods. Across three benchmark evaluations, MetaAPO consistently demonstrates superior performance.
> - **Different datasets.** Our main experiments are conducted on UltraFeedback, a dataset widely used in recent alignment research. In addition, Table 6 reports results on additional alignment datasets. We also ran experiments under the SimPO-like self-generated data setting mentioned by the reviewer, further confirming the robustness and stability of MetaAPO.
> - **Different reward models.** To ensure experimental rigor, we train reward models on UltraFeedback using the corresponding base model, thereby avoiding any preference leakage. We also compare performance using open-source reward models in Table 5 of the appendix.
> - **Different numbers of generated rollouts.** Figure 4 (Left) evaluates how the number of generated online rollouts (K) affects performance. MetaAPO remains stable across different rollout sizes and consistently outperforms competing baseline algorithms.

---

> > ### Author Response · Authors · 2025-11-28
> > **Responses to Reviewer Xifs (Part 3)**
> >
> > We appreciate the reviewers' suggestions on the rigor of our paper's presentation. Below are explanations for the specific issues:
> >
> > > Q1: Explanation for “all samples”
> >
> > When plotting the scatter plot in Figure 2 (Middle), we randomly sampled 600 out of approximately 20,000 samples for visualization purposes, as not all sample points could be accommodated in the display. For Figure 3 (Right), we used all ~20,000 samples to ensure the accuracy of the overall distribution. Both come from the same distribution, and we did not perform any specific selection on the data.
> >
> > > Q2: Explanation for positive values in Figure 3 (Right)
> >
> > Yes, $\log\sigma(\cdot)$ is always negative. Since the smoothing of KDE may result in values greater than 0, we have updated Figure 3 by adjusting the KDE to directly use histograms to represent the distribution.
> >
> > > Q3: Explanation for notation in Figure 2 (Middle)
> >
> > In **Figure 2 (Middle)**, we aim to clearly separate the positive and negative regions of the x-axis to better illustrate the meta-learner’s sampling behavior. Therefore, We replace the original preference score $\ell(\cdot)=\log\sigma(\beta\cdot m(\cdot))$ with the implicit reward margin $m(\cdot)=\log \frac{\pi_{\theta}(y_w \mid x)}{\pi_{\text{ref}}(y_w \mid x)} - \log \frac{\pi_{\theta}(y_l \mid x)}{\pi_{\text{ref}}(y_l \mid x)}$. This rescaling only changes the numerical magnitude of the values and does not affect their relative ordering or the qualitative sampling trends. We also confirmed that the original score $\ell(\cdot)$ exhibits a similar near-linear negative relationship, which is consistent with the pattern by $m(\cdot)$. We have updated Figure 2 and its accompanying explanation in the paper to present this more clearly.
> >
> > In response to the reviewer’s request, we report the statistics of the samples used in Figure 2 (Middle). Specifically:
> >
> > - **Offline margin** $m^{\text{off}}(\cdot)$**:** Mean = 1.24, Var = 181.84
> > - **Online margin** $m^{\text{on}}(\cdot)$**:** Mean = 2.80, Var = 24.57
> >
> > In **Figure 3 (Right)**, we visualize the probability densities of the offline and online implicit reward margins $m(\cdot)$. The results show that the two distributions exhibit overlap in their value ranges.
> >
> > > Q4: More behavior of meta-learner
> >
> > Further analysis of the results obtained under the settings where SimPO score is used without warmup reveals some phenomena that further illustrate the dynamic nature and robustness of the meta-learner.
> >
> > **(1) In offline Ultrafeedback dataset (w/ SimPO score)**
> >
> > We observed that when calculating preference scores directly using SimPO on the offline dataset (without DPO warmup), the meta-learner automatically sampled more data for online enhancement (**67%** sampling rate) compared to the standard DPO warmup method (58%). Additionally, the shape of the meta-learner's weighting function became **steeper** (squeezed, resembling a sharper sigmoid).
> >
> > - **Interpretation:** Without warmup, the distribution mismatch between the policy and the offline data is more pronounced. The meta-learner identifies a larger "alignment gap", prompting a more aggressive strategy: it triggers online exploration more frequently to bridge this gap, resulting in a higher sampling rate.
> >
> > **(2) In self-generated Ultrafeedback dataset (SimPO-like setting)**
> >
> > Conversely, when training on the self-generated dataset (where offline data is replaced by model-generated samples), the distribution gap is smaller. In this setting, we observed that the quantity of sampled data **decreased to 55%**, and the meta-learner's function became **smoother** (flatter).
> >
> > - **Interpretation:** Since the offline data here is generated by the model itself, it is closer to on-policy data, indicating the OOD gap is smaller ($m^{\text{off}}(\cdot)$: Mean = 1.47, Var = 17.39; $m^{\text{on}}(\cdot)$: Mean = 1.83, Var = 11.29). The meta-learner detects that the potential marginal gain from further online exploration is lower compared to the standard offline setting. Consequently, it adopts a more conservative strategy, placing more trust in the existing data and reducing unnecessary online sampling.
> >
> > These contrasting behaviors confirm that the meta-learner is not a static filter but a dynamic regulator that adjusts the exploration-exploitation trade-off based on the severity of the distribution mismatch.
> >
> > We sincerely thank you for your time and for the thoughtful evaluation of our work. We are very grateful for your constructive feedback throughout this process. We will include these discussions and acknowledgements for the reviewers' contributions to improving the quality and depth of our paper in the revised version. We hope that these efforts address the concerns raised and provide sufficient justification for reconsidering the overall assessment.
> >
> > If you have further questions, feel free to let us know.

---

### Official Review · Reviewer_j3Qa · 2025-10-31

**Soundness:** 3
**Presentation:** 4
**Contribution:** 3
**Rating:** 8
**Confidence:** 4

**Summary:**

This paper proposes MetaAPO, a meta-learning-based framework for adaptive preference optimization. Instead of statically mixing offline and online preference data, the method introduces a meta-learner that dynamically estimates the alignment gap between the current policy and offline data. The meta-learner assigns sample-specific weights $w_i$ and decides whether to perform online augmentation. This yields an adaptive data selection and loss weighting mechanism that focuses computation on high-gain samples. Experiments on Llama-3.1-8B and Qwen2.5-7B across AlpacaEval 2, Arena-Hard, and MT-Bench show consistent improvements over offline (DPO, IPO, KTO), online (PPO, Online DPO), and hybrid (SELM, ADPO) baselines.

**Strengths:**

1. The paper introduces a clear and well-motivated framework that integrates meta-learning into preference optimization. By learning sample-specific weights to balance offline and online data, MetaAPO effectively unifies filtering and augmentation within a single adaptive process, avoiding the heuristic thresholds used in prior methods such as ADPO or Selective DPO.
2. The proposed meta-weighted sampling mechanism demonstrates strong empirical advantages. It focuses training resources on high-gain samples, achieves smoother “explore–integrate” dynamics, and significantly reduces annotation costs while maintaining or improving performance across multiple benchmarks.
3. The experimental evaluation is comprehensive. The authors include diverse baselines (offline, online, and hybrid), three major alignment benchmarks and detailed ablations on sampling and loss weighting.

**Weaknesses:**

One of the limitation is the sensitivity to hyperparameters introduced by the meta-learner, particularly the update interval $T_{meta}$. Achieving optimal results appears to require careful tuning, which may increase the implementation burden. However, this does not substantially detract from the overall contribution given the consistent empirical gains.

**Questions:**

None

---

> ### Author Response · Authors · 2025-11-17
> **Responses to Reviewer j3Qa**
>
> Thank you for your thoughtful and encouraging feedback. We sincerely appreciate your positive assessment of MetaAPO’s motivation and experimental design.
>
> Regarding the sensitivity to meta-learner update interval $T_\textbf{meta}$, we provide both theoretical and empirical support. In Theorem 1, we demonstrate that a large interval can lead to stable generalization, and Figure 3 (Left) experimentally confirms that performance remains robust across a range of intervals. We will explore more efficient implementations in the future.
>
> Thank you again for your constructive and supportive review.

---

### Official Review · Reviewer_X3TX · 2025-10-31

**Soundness:** 3
**Presentation:** 2
**Contribution:** 2
**Rating:** 4
**Confidence:** 3

**Summary:**

This paper proposes MetaAPO, which introduces a lightweight meta-learner into preference optimization (RLHF/DPO). The meta-learner dynamically learns sample weights from offline preference scores to jointly guide online sample selection and the weighting of offline/online data during training. Across several mainstream benchmarks, MetaAPO outperforms existing alignment methods (e.g., DPO, IPO), achieving better alignment performance and higher data efficiency.

**Strengths:**

(1) Proposes a clear, intuitive, and novel online–offline joint optimization framework with strong practical value.
(2) Demonstrates consistent effectiveness across multiple models and benchmarks.
(3) Provides solid theoretical derivations with comprehensive ablation and robustness studies.

**Weaknesses:**

(1) The meta-learner’s input features are overly simplistic, which may limit generalization. Although the appendix briefly explores multi-feature inputs (Table 7), the main experiments rely solely on a single offline score, lacking analysis of why this is sufficient.
(2) The paper lacks qualitative and human evaluation. Results rely solely on GPT-4o as the judge, without validating consistency with human preferences. Case analyses are limited in number and do not cover complex scenarios (e.g., multi-turn dialogue or safety alignment).
(3) The online sampling design (K=8 and stochastic thresholding) is not empirically justified, and its sensitivity to these parameters remains unexplored.
(4) Recent baselines (e.g., DPO-Shift, MAP) and multi-stage alignment frameworks (e.g., RLHF + DPO hybrids) are not included, which may under- or overestimate MetaAPO’s relative advantages.

**Questions:**

(1) Have you tried adding richer features (e.g., model gradients, response semantic similarity) as inputs to the meta-learner? If not, please discuss potential information loss from relying solely on a single offline score.
(2) Could you report results for different numbers of sampled candidates (K=4/12/16) to analyze the trade-off between performance and computational cost?
(3) Can you include a small-scale human evaluation or justify the consistency between GPT-4o judgments and human preferences?

---

> ### Author Response · Authors · 2025-11-17
> **Responses to Reviewer X3TX (Part 1)**
>
> We appreciate your great efforts in reviewing our paper and hope that the following responses can address all of your concerns.
>
> > Q1: Richer features for meta-learner input
>
> **(1) Experiments with semantic similarity as additional input features**
>
> In response to the reviewer’s suggestion, we conducted experiments incorporating semantic similarity as an additional input to the meta-learner. Specifically, we used Qwen/Qwen3-Embedding-0.6B to compute embeddings for the chosen and rejected responses in the offline dataset. We then computed their cosine similarity and fed this feature with the original offline preference score $\ell$ into the meta-learner for weight prediction, which then drives both sample selection, model optimization and meta-learner updates.
>
> The results are shown in the table below. We find that adding semantic similarity leads to **comparable performance**, indicating that the original preference score already provides sufficient information for meta-learner.
>
> |  | **AlpacaEval2 WR** | **AlpacaEval2 LR** | **Arena-Hard WR** | **Arena-Hard SC** | **MT-Bench** |
> | --- | --- | --- | --- | --- | --- |
> | MetaAPO | 47.48 | 43.21 | 43.9 | 40.8 | 7.56 |
> | MetaAPO (w/ Semantic Similarity) | 47.83 | 42.88 | 43.7 | 40.5 | 7.50 |
>
> **(2) Why the offline preference score is a sufficient input**
>
> - **Dynamic and model-dependent signal.** The offline preference score **directly** measures how well the current model aligns with offline human preferences. It captures (i) model-specific alignment states and (ii) sample-specific information. As model parameters evolve throughout training, the preference score also changes dynamically, providing a rich and continuously updated signal for estimating the alignment gap.
> - **Strong empirical correlation with online gain.** As shown in **Figure 2 (Middle)**, our empirical analyses show the offline preference score $\ell_\text{off}$ exhibits a clear correlation with the online–offline score gap ($\ell_\text{on} - \ell_\text{off}$). This correlation allows a single-scalar-input meta-learner to reliably function as an “alignment gap estimator”, identifying samples with high potential gains from online augmentation.
> - **Simplicity and efficiency**. The preference score is readily available from the alignment loss without additional computation or auxiliary models. In contrast, semantic similarity features require additional embedding models and inference. **Our primary goal is to demonstrate the effectiveness of the *dynamic and model-aware* sampling mechanism for mitigating OOD issues during alignment.**
>
> We have added these results and discussions in **Table 2 (Section 5.4) and Appendix C.5** in the revised version and will further explore richer feature designs as an interesting direction for future work.
>
> > Q2: Effect of candidate sampling size (K) and sampling thresholds
>
> **(1) Candidate sampling size (K)**
>
> For all baselines that rely on online sampling, we use the same number of candidates (K = 8) to ensure a fair comparison. To further analyze the sensitivity of MetaAPO to the choice of K, we conducted additional experiments on Llama-3.1-8B with different values of K (4, 12, 16). As summarized in the Table below, the results show that MetaAPO remains robust within this range. Notably, even with a small number of online candidates (K=4), MetaAPO still outperforms the baselines, indicating that the improvements do not rely on a particular choice of K. The Annotation Ratio is calculated relative to Online DPO with K=8.
>
> |  | **AlpacaEval2 WR** | **AlpacaEval2 LR** | **Annotation Ratio(%)** |
> | --- | --- | --- | --- |
> | Online DPO (K=4) | 42.26 | 38.19 | 50 |
> | Online DPO (K=8) | 43.75 | 39.98 | 100 |
> | Online DPO (K=12) | 43.79 | 38.99 | 150 |
> | Online DPO (K=16) | 44.28 | 40.11 | 200 |
> | MetaAPO (K=4) | 46.43 | 42.29 | 31 |
> | MetaAPO (K=8) | 47.48 | 43.21 | 58 |
> | MetaAPO (K=12) | 48.06 | 44.75 | 89 |
> | MetaAPO (K=16) | 48.58 | 44.41 | 114 |
>
> We have added a visualization of these results in **Figure 4 (Left)** of the revised manuscript to better illustrate the performance trend as K varies.
>
> **(2) Sampling thresholds**
>
> In our ablation study, we evaluated several alternative sampling strategies, including random sampling, fixed-threshold sampling and sampling based on a fixed sigmoid function.
>
> To further isolate the effect of adaptivity, we conducted an additional experiment where we **initialized a new meta-learner with the weights learned by MetaAPO, froze it without any updates,** and used this fixed meta-learner to sample within MetaAPO. As shown in Table below, the fixed sampling strategies lead to performance decline. These results consistently demonstrate the necessity of using a dynamic sampling mechanism.
>
> |  | **AlpacaEval2 WR** | **AlpacaEval2 LR** | **Arena-Hard WR** | **Arena-Hard SC** | **MT-Bench** |
> | --- | --- | --- | --- | --- | --- |
> | MetaAPO | 47.48 | 43.21 | 43.9 | 40.8 | 7.56 |
> | w/ Fixed Sampling | 45.90 | 42.04 | 40.2 | 37.1 | 7.34 |

---

> ### Author Response · Authors · 2025-11-17
> **Responses to Reviewer X3TX (Part 2)**
>
> > Q3: Evaluation under human judgement
> >
>
> **(1) Human annotation and ranking consistency**
>
> To further validate the reliability of our evaluation, we conducted a small-scale human study using **50 randomly sampled prompts and responses from AlpacaEval2**. Human annotator compared each model’s response against the same reference model (**GPT-4o-2024-05-13**),  following the LLM-Arena protocol—i.e., anonymizing model identities and presenting paired responses in randomized order to ensure fairness and reduce bias.
>
> MetaAPO still outperforms other baseline methods in human evaluations. The results are summarized below:
>
> |  | **Win (%)** | **Tie (%)** | **Loss (%)** |
> | --- | --- | --- | --- |
> | Online DPO | 42 | 10 | 48 |
> | SELM | 34 | 4 | 62 |
> | PPO | 42 | 8 | 50 |
> | MetaAPO | 48 | 12 | 40 |
>
> Please refer to **Figure 4 (Right)** in the revised manuscript for a visualization of these results.
>
> - **Qualitative findings**
>
> Human annotators reported that MetaAPO’s outputs are often **as strong as or sometimes better** than those of GPT-4o-2024-05-13 in alignment-related dimensions such as helpfulness, coherence, safety awareness, and stylistic naturalness.
>
> We have added more detailed examples in **Figures 8, 9, 10 (Appendix D)**, including multi-turn conversation cases and safety-related prompts.
>
> **(2) Additional ranking consistency with multiple AI judges**
>
> Beyond the human study, we evaluated methods using two additional independent judges: **DeepSeek-V3.1-Terminus** and **GPT-5**. Across these different evaluators, we observed fluctuations in absolute win-rate values, but **the relative ranking among models remained fully consistent**, which is consistent with the stability characteristics reported in the AlpacaEval2 paper.
>
> In all settings, MetaAPO consistently **outperformed** Online DPO, SELM and PPO, demonstrating that the performance gains of MetaAPO are robust across both human judgments and diverse automated evaluators.
>
> |  | **AlpacaEval 2 WR** | **AlpacaEval 2 LR** |
> | --- | --- | --- |
> | Online DPO **(Deepseek-V3.1-Terminus)** | 39.32 | 31.46 |
> | SELM | 38.56 | 31.73 |
> | PPO | 42.71 | 32.81 |
> | MetaAPO | 45.82 | 35.89 |
> | Online DPO **(GPT-5)** | 35.16 | 22.54 |
> | SELM | 32.32 | 20.60 |
> | PPO | 35.66 | 22.41 |
> | MetaAPO | 40.30 | 25.32 |
>
> > Q4: Comparison with more recent baselines
> >
>
> We further incorporated several recent baseline methods into our comparison to provide a more comprehensive comparison under the online sampling setting. The results are shown in the table below.
>
> Specifically, **DPO-Shift** [1] introduces a re-weighting strategy for negative samples but shows only marginal improvements compared with DPO. We also added **BeeS** [2] and **MAP** [3], both of which enhance preference optimization by selecting or filtering higher-quality data using their respective sampling heuristics.
>
> For all baselines, we used the **default thresholds and hyperparameters** reported in their original papers to ensure faithful reproduction. A detailed description of these baselines and our implementation has been added to **Appendix B.3**.
>
> Across all metrics, MetaAPO consistently outperforms these baselines. Please refer to **Table 1** for complete results.
>
> |  | **AlpacaEval2 WR** | **AlpacaEval2 LR** | **Arena-Hard WR** | **Arena-Hard SC** | **MT-Bench** |
> | --- | --- | --- | --- | --- | --- |
> | MetaAPO | 47.48 | 43.21 | 43.9 | 40.8 | 7.56 |
> | DPO-Shift | 19.33 | 19.39 | 28.7 | 27.6 | 7.03 |
> | BeeS | 41.42 | 37.45 | 39.5 | 36.4 | 7.13 |
> | MAP | 43.60 | 38.97 | 40.4 | 37.2 | 7.17 |
>
> References:
>
> [1] Yang et al. DPO-Shift: Shifting the Distribution of Direct Preference Optimization, Arxiv, 2025.
>
> [2] Deng et al. Less is more: Improving llm alignment via preference data selection, NeurIPS, 2025.
>
> [3] Huang et al. Larger or Smaller Reward Margins to Select Preferences for LLM Alignment?, ICML, 2025.

---

> ### Author Response · Authors · 2025-11-24
>
> Dear Reviewer X3TX,
>
> We greatly appreciate the time and effort you've taken to review our submission. We hope that our response has addressed the concerns raised in your initial reviews, and we look forward to your feedback.
>
> Please let us know if you need any further information or clarification.
>
> Best regards and thanks,
>
> Paper 12533 Authors

---

### Author Response · Authors · 2025-11-17
**Revision Uploaded**

We thank all reviewers for the constructive comments and have updated our paper accordingly. Please take a moment to check out the revised manuscript, which incorporates the following key modifications (highlighted by blue):

**New Results:**

1. Added new baseline models: DPO-Shift [1], BeeS [2] and MAP [3] (Table 1, Section 5).
2. Added experiments with additional semantic similarity as meta-learner input (Table 2, Section 5.4).
3. Added ablation study with fixed learned meta-learner to predict sampling weight and loss weight (Table 2, Section 5.4).
4. Conducted a sensitivity analysis and provided visualizations of the sampling size K (Figures 4 (Left), Appendix C.5).

**Other Revisions:**

1. Clarified the difference between “All Sampling” in ablation study and Online DPO (Section 5.4)
2. Added the introduction of new baseline models: DPO-Shift [1], BeeS [2] and MAP [3] (Appendix B.3)
3. Added human evaluation results for model response win rates (Figures 4 (Right), Appendix C.5)
4. Added more case studies about MetaAPO responses about safety and multi-turn (Figure 8, 9, 10 , Appendix D)

We appreciate the great efforts by the AC and reviewers. Please do not hesitate to let us know if you have any additional comments on the paper.

References:

[1] Yang et al. DPO-Shift: Shifting the Distribution of Direct Preference Optimization, Arxiv, 2025.

[2] Deng et al. Less is more: Improving llm alignment via preference data selection, NeurIPS, 2025.

[3] Huang et al. Larger or Smaller Reward Margins to Select Preferences for LLM Alignment?, ICML, 2025.

---

### Author Response · Authors · 2025-12-01
**Summary of Rebuttal**

We thank the Area Chair and all reviewers for their valuable feedback, which has improved the rigor and clarity of our work. We are encouraged by the recognition of our work as **"well-motivated"** (Reviewer j3Qa, Xifs) and **"strong practical value with solid theoretical derivations"** (Reviewer X3TX), with **"strong empirical advantages"** (Reviewer j3Qa) and **"demonstrate the effectiveness over existing methods"** (Reviewer Xifs). Here is the summary of discussion points during the rebuttal period:

**Concerns & Responses**

- **Reviewer X3TX**. We addressed all raised questions and integrated the requested experiments into the revised manuscript:
    - **Q1 (Richer Features):** Evaluated semantic similarity as an additional meta-learner input. Results confirm that the preference score sufficiently captures dynamic model-aware alignment states and sample-specific information (**Table 2**).
    - **Q2 (Human Evaluation):** Conducted a human study and cross-validated with multiple LLM judges. Evaluations exhibit high consistency and show that MetaAPO achieves SOTA performance (**Figure 4**).
    - **Q3 (Sampling Size):** Added experiments with varied candidate sizes (K = 4/12/16), showing MetaAPO remains robust (**Figure 4**).
    - **Q4 (New Baselines):** Incorporated comparisons with DPO-Shift, MAP, BeeS, where MetaAPO maintains SOTA (**Table 1**).
- **Reviewer j3Qa**
    - **The reviewer expressed strong support**. Regarding the comment on the hyperparameter (meta-learner update interval), the reviewer noted that this "does not substantially detract from the overall contribution given the consistent empirical gains." We further supported this with **Theorem 1** and ablation studies in the revision.
- **Reviewer Xifs** *(Note: We provided detailed clarifications for each concern. While the reviewer confirmed that the fixed weighting concern was resolved, the discussion period ended before they could reply to our final results regarding warmup and SimPO settings.)*
    - **Q1 (Mechanism Rationality):** We noted a misunderstanding regarding our mechanism and clarified that our approach aligns with the reviewer's intuition to focus on low-score (misaligned) samples.
    - **Q2 (Ablation Study**): Added additional ablation studies showing that fixed Sampling/Loss Weighting consistently underperforms the dynamic MetaAPO (Table 2). We also included a visualization analysis of the dynamic adaptation process (Figure 3 Middle). **The reviewer explicitly confirmed that this concern has been addressed.**
    - **Q3 (Generalization to SimPO setting):** Conducted specific experiments in a SimPO-like setting (using self-generated data as offline data). MetaAPO continued to outperform baselines, demonstrating that **MetaAPO imposes no constraints on the source of offline data and significantly extending the applicability.**
    - **Q4 (Warmup Dependency):** We demonstrated that MetaAPO achieves comparable performance with reduced or even no warmup. This eliminates concerns regarding artifacts from offline data and proves the method's robustness.
    - **Q5 (Presentation):** We improved the clarity of definitions in ablation studies and visualizations in Figures 2 & 3.
    - **Q6 (New Baselines):** We included recent methods (DPO-Shift, MAP, BeeS), and MetaAPO maintained SOTA performance.

We have incorporated the clarifications, new analyses and updated results into the revision. We believe the additional evidence and explanations comprehensively address the reviewers’ concerns and further strengthen the contribution of MetaAPO.

---

### Meta-Review · Area_Chair_bK7a · 2026-01-07

**Summary:**

The main concerns affecting the decision:
- insufficiently justified designs in the meta-learner: input features, objective, and sampling/weighting behavior of meta-learner. (X3TX, j3Qa, Xifs)
- missing or incomplete ablations/baselines: needed to isolate what actually provides the benefit (e.g., comparison to fixed schemes derived from the learned mapping; newer baselines and filtering methods). (X3TX, j3Qa, Xifs)
- evaluation gaps: no human/qualitative evaluation; reliance on GPT-4o as judge; limited coverage of multi-turn/safety settings. (X3TX)
- uncertain reasonability and generality: whether the gains are robust and whether the approach generalizes to common settings. (Xifs)
- unclear and sometimes inconsistent settings/notation/analysis that undermines confidence in the claimed mechanism (especially online-offline data mixing, warm-up training). (Xifs)
- incomplete literature review: some recent studies are missing. (Xifs)

**Reviewer Concerns:**

Addressed by the rebuttal (partially):
- meta-learner designs: additional experiments are done. (X3TX, j3Qa, Xifs)
- experiments/ablations: the revision already includes broad benchmark coverage, ablations, and hyperparameter sensitivity. (j3Qa, Xifs)
- evaluation gaps: human evaluations are added. (X3TX)
- inconsistent settings: some experiments are added. (Xifs)

Still outstanding:
- reasonability of the meta-learner objective/weighting behavior: the reviewer remains unconvinced that emphasizing high-score samples and downweighting low-score samples is principled in general. The explanation about avoiding redundant sampling and mitigating OOD for low-score samples does not resolve the core question. (Xifs, major)
- generality of the pipeline: the approach appears to rely on a warm-up/preconditioning stage on offline data, which could bake information into the preference scores and make the meta-learner’s decisions less generally applicable. Whether MetaAPO still helps without such preconditioning, or when offline/online are closer, remains unclear. (Xifs, major)
- unclear and inconsistent analysis/notation/interpretability (Xifs, major): these reduce trust in the mechanistic story and the experimental design. (Xifs, major)

**Reviewer Scores:**

- X3TX (original score 4): likely small upward improvements, as major requests appear fully satisfied.
- j3Qa (original score 8): likely no change, which is already strongly supportive; however, such comments are too oversimplistic to support.
- Xifs (original score 4): likely small upward improvements

---

### Decision · Program_Chairs · 2026-01-26

Accept (Poster)